# ON THE EVOLUTION OF LANGUAGE MODELS WITHOUT LABELS: *Majority Drives Selection, Novelty Promotes Variation*

## ABSTRACT

Large language models (LLMs) are increasingly trained with reinforcement learning from verifiable rewards (RLVR), yet real-world deployment demands models that can self-improve without labels or external judges. Existing self-improvement approaches primarily rely on self-confirmation signals (e.g., confidence, entropy, or consistency) to generate rewards. This reliance drives models toward over-confident, majority-favored solutions, causing an entropy collapse that degrades pass@$n$ and reasoning complexity. To address this, we propose EVOL-RL, a label-free framework that mirrors the evolutionary principle of balancing selection with variation. Concretely, EVOL-RL retains the majority-voted answer as an anchor for stability, but adds a novelty-aware reward that scores each sampled solution by how different its reasoning is from other concurrently generated responses. This *majority-for-stability* + *novelty-for-exploration* rule mirrors the variation–selection principle: *selection prevents drift, while novelty prevents collapse*. Evaluation results show that EVOL-RL consistently outperforms the majority-only baseline; e.g., training on label-free AIME24 lifts Qwen3-4B-Base AIME25 pass@1 from baseline's 4.6% to 16.4%, and pass@16 from 18.5% to 37.9%. EVOL-RL not only prevents in-domain diversity collapse but also improves out-of-domain generalization (from math reasoning to broader tasks, e.g., GPQA, MMLU-Pro, and BBEH).

## 1 INTRODUCTION

The reasoning capabilities of Large Language Models (LLMs) have advanced dramatically, particularly through paradigms like Reinforcement Learning with Verifiable Rewards (RLVR) (Jaech et al., 2024; Guo et al., 2025; Yang et al., 2025). The next frontier of intelligence lies in enabling LLMs to autonomously evolve, continuously learning from the vast, unlabeled data streams they encounter in real-world environments. This *label-free evolving* paradigm allows a model to iteratively improve itself while solving tasks, without relying on ground-truth labels or external judges, making it both practical and necessary. However, turning inference into learning reopens a long-standing RL problem: balancing exploration and exploitation. This dilemma becomes especially severe in label-free settings, where models must rely on internal signals (e.g., inherent self-consistency, entropy, or confidence) to generate rewards for themselves (Grandvalet & Bengio, 2004; Lee et al., 2013; Zuo et al., 2025; Shafayat et al., 2025; Li et al., 2025b).

The fundamental flaw in relying on internal signals is not merely that they are initially noisy or biased, but that the learning process itself actively degrades the quality of the reward signal over time (Liang et al., 2024; Zeng et al., 2024). By rewarding conformity to its self-confirmation, the model systematically eliminates the solution diversity (Lee et al., 2024). This creates a degenerative feedback loop: a progressively narrower and more biased policy generates an increasingly impoverished reward signal, which in turn accelerates the policy's collapse into a low-entropy state (Ding et al., 2025; Liang et al., 2024). Similar dynamics are well known in RL and self-training when entropy regularization or external supervision is absent (Haarnoja et al., 2018). Recent studies also show that training on self-generated data can harm diversity over time (Shumailov et al., 2024) and eventually lead to collapse. Figure 1 illustrates this phenomenon in reasoning: under traditional Test-Time Reinforcement Learning (TTRL) (Zuo et al., 2025), pass@1 may rise but pass@$n$ drops, while response length and complexity steadily decline, indicating that the model fails to evolve.

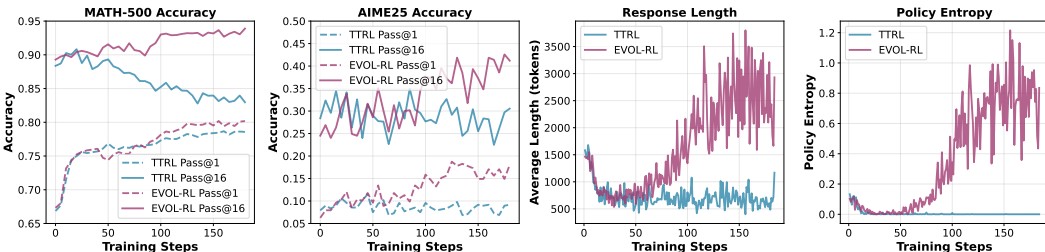

Figure 1: TTRL's entropy collapse vs. EVOL-RL's diversity preservation on Qwen3-4B-Base (trained label-free on MATH-500). Majority-only TTRL drives pass@$n > 1$ down, shortens reasoning, and collapses entropy, whereas EVOL-RL improves accuracy, sustains reasoning diversity.

In this paper, we ground LLM evolving in the simple rule behind biological evolution: ***variation*** creates new candidates; ***selection*** keeps what works. Existing methods effectively implement only the ***selection*** half of evolution, driving the population toward whatever the model already believes. This majority-only (or entropy minimization, confidence maximization) reinforcement amplifies existing biases and often leads entropy collapse and shrinking response diversity, as shown above. Our formulation restores the full evolutionary loop: we pair ***selection***, which stabilizes optimization by keeping high-quality solutions, with ***variation***, which explicitly promotes novelty and sustains exploration. This idea is deeply rooted in decades of evolutionary computation research, including genetic algorithms (Holland, 1992; Eiben & Smith, 2015), novelty search (Lehman & Stanley, 2011), and quality–diversity (QD) methods such as MAP-Elites (Pugh et al., 2016), which collectively show that diversity preservation is essential for avoiding collapse and enabling robust, long-term progress.

Hence, we propose ***EVolution-Oriented and Label-free Reinforcement Learning (*EVOL-RL*)***, a simple objective that combines a stabilizing ***selection*** signal with an explicit ***variation*** incentive. Concretely, EVOL-RL retains the majority-voted answer as the anchor for stability, but adds a novelty-aware reward that scores each sampled solution by how different its reasoning is from other concurrently generated responses (semantic similarity of their reasoning traces). This majority-for-stability + novelty-for-exploration rule mirrors the variation–selection principle: *selection prevents drift; novelty prevents collapse*. As demonstrated in Figure 1, EVOL-RL successfully averts all symptoms of diversity collapse, fostering a healthy equilibrium between refining known solutions and discovering new ones. This balanced approach translates into substantial performance gains, especially in out-of-domain generalization. For instance, after training on AIME24, EVOL-RL elevates the Qwen3-4B-base model's pass@1 accuracy on the AIME25 benchmark from 4.6% (TTRL) to 16.4%, while more than doubling the pass@16 accuracy from 18.5% to 37.9%.

**Contributions.** (1) We diagnose why majority-only objectives shrink exploration during label-free training and formalize their link to entropy collapse on reasoning tasks. (2) We provide a new perspective on label-free learning by framing it as an evolutionary system. This view allows us to diagnose diversity collapse as a form of premature convergence and solve it by applying the core evolutionary principle of balancing selection with variation. (3) We design a practical novelty-aware reward that complements majority selection and enables stable, label-free improvement. Across math benchmarks, EVOL-RL reverses the pass@$n$ decline, maintains longer and more informative chains of thought, and improves out-of-domain accuracy, while remaining simple to implement. (4) We deliver state-of-the-art results in unsupervised RL training, demonstrating that EVOL-RL achieves significant out-of-domain generalization gains where prior methods fail, such as more than tripling pass@1 accuracy and doubling pass@16 accuracy on AIME25 benchmark. (5) We provide a theoretical analysis of entropy stabilization in Appendix D. We formally prove that while a correctness-only objective allows the optimal policy to collapse onto a single response, our novelty-augmented objective *guarantees* that the optimal policy must distribute probability mass across multiple correct modes, providing a rigorous foundation for the method's stability.

## 2 RELATED WORK

**Enhancing Reasoning in Large Language Models.** Significant progress in LLM reasoning has been driven by RLVR (Jaech et al., 2024; Guo et al., 2025; Yang et al., 2025; Yu et al., 2025; Xiong et al., 2025; Dai et al., 2025b), which fine-tunes models using RL on tasks where an automated

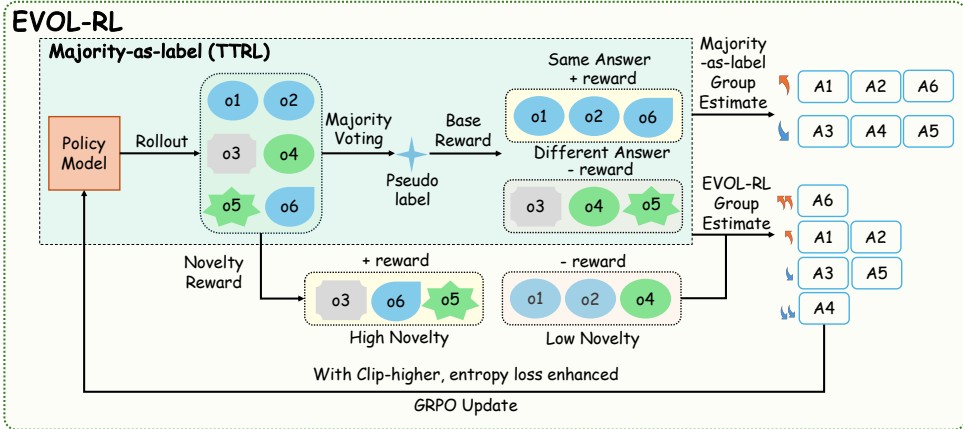

Figure 2: An overview of the EVOL-RL framework. For each prompt, the policy generates multiple responses. These are grouped by their final answer to identify the majority group. A novelty score is then computed for each response based on its semantic dissimilarity to others. Finally, a reward is assigned based on both majority (selection) and novelty (variation), guiding the policy update via GRPO. In the illustration, colors group responses by their final answer, while different marker shapes indicate semantically distinct reasoning paths.

verifier can confirm the correctness of the final answer, such as mathematics and coding (Zeng et al., 2025; Wang et al., 2025a;b; Cui et al., 2025; Huang et al., 2025; Dai et al., 2025a; Zheng et al., 2025b; Zhou et al., 2025b; Zheng et al., 2025a; Fang et al., 2025). While highly effective, the reliance of RLVR on external verifiers restricts its applicability to domains with deterministic, easily checkable solutions (Zhao et al., 2025c;a; Zhou et al., 2025a; 2024). Our work contributes to the effort of improving reasoning in more general domains where such verifiers are unavailable.

**Label-Free Adaptation and Self-Improvement.** To overcome the limitations of verifiers and adapt to new data distributions, researchers have focused on label-free learning methods that generate reward signals without ground-truth labels. These approaches primarily fall into two categories. One line of research derives rewards from the model's intrinsic confidence, training the model to become more "certain" by rewarding low-entropy or self-consistent outputs (Prabhudesai et al., 2025; Agarwal et al., 2025; Zhao et al., 2025b; Zhang et al., 2025b; Shafayat et al., 2025; Chung et al., 2025; Li et al., 2025a; Zhang et al., 2025a). The other prominent paradigm, which our work directly addresses, bootstraps supervision from majority. Test-Time Reinforcement Learning (TTRL) exemplifies this by using the majority-voted answer from multiple samples as a pseudo-label for RL updates (Zuo et al., 2025). While empirically powerful, we identify a critical flaw in the majority-driven approach: it suppresses solution diversity and actively punishes correct but non-mainstream reasoning, leading to the entropy collapse we describe. As demonstrated in the ablation study (Sec. 4.3), simply adding entropy loss and clip-high (Cui et al., 2025; Shen, 2025; Park et al., 2025) cannot overcome this issue.

Given that such non-directional exploration is insufficient, our work is the first to pin down the "majority trap" and demonstrate that instead of a generic entropy bonus, a directional novelty reward that re-ranks credit based on semantic uniqueness is required. Our approach is thus fully label-free and redesigns the reward signal itself. This is fundamentally distinct from methods that operate in supervised RLVR settings (Dai et al., 2025a), require separate trained evaluators (Li et al., 2025a; Pang et al., 2023), attempt to stabilize training via curriculum learning (Roy et al., 2025), or simply adjust exploration within the old self-consistency framing (Liu et al., 2025). This coupling of the majority signal with directional, population-relative novelty is the key to preventing collapse.

## 3 METHOD

Our approach is illustrated in Figure 2. which uses Group Relative Policy Optimization (GRPO) (Shao et al., 2024) as its optimization algorithm, but guides it with a novel reward function that explicitly balances majority with novelty.

## 3.1 Optimization with GRPO

GRPO is a policy-gradient algorithm designed for fine-tuning LLMs without a separate value function. Its central idea is to evaluate each sampled response relative to a group of its peers generated for the same prompt. This relative evaluation is then used to update the policy with a PPO-style clipped objective, regularized by a KL penalty to ensure stable learning.

For a given prompt $\mathbf{q}$, a policy LLM $\pi_{\theta_{\text{old}}}$ generates a group of $G$ complete responses $\{\mathbf{o}_1, \ldots, \mathbf{o}_G\}$. Each response $\mathbf{o}_i$ receives a scalar reward $r_i$. Rewards within the group are normalized with a z-score to obtain a response-level advantage:

$$\hat{A}_i = \frac{r_i - \text{mean}(r_1, \ldots, r_G)}{\text{std}(r_1, \ldots, r_G)},$$

The policy is optimized with a clipped surrogate objective:

$$\frac{1}{G} \sum_{i=1}^{G} \frac{1}{|o_i|} \sum_{t=1}^{|o_i|} \min\left\{ \frac{\pi_\theta(o_{i,t} \mid \mathbf{q}, \mathbf{o}_{i,<t})}{\pi_{\theta_{\text{old}}}(o_{i,t} \mid \mathbf{q}, \mathbf{o}_{i,<t})} \hat{A}_{i,t}, \ \text{clip}\left( \frac{\pi_\theta(o_{i,t} \mid \mathbf{q}, \mathbf{o}_{i,<t})}{\pi_{\theta_{\text{old}}}(o_{i,t} \mid \mathbf{q}, \mathbf{o}_{i,<t})}, 1 - \epsilon_{\text{low}}, 1 + \epsilon_{\text{high}} \right) \hat{A}_{i,t} \right\} \tag{1}$$

## 3.2 Reward Design: Implementing Selection and Variation

Our reward design directly implements the principles of selection and variation to counteract diversity collapse. Selection, based on correctness via majority vote, provides a stable signal to prevent the policy from drifting. Variation, driven by semantic novelty, provides the exploratory pressure needed to maintain a diverse set of reasoning strategies.

A key design choice is that the novelty incentive is applied strategically to all solutions—both those that agree with the majority and those that do not. For majority-aligned solutions, rewarding novelty encourages the model to discover multiple valid reasoning paths to the correct answer, directly fighting the decline in pass@n performance. For minority solutions, rewarding novelty is crucial for escaping local optima. It discourages policy collapse into a few high-frequency failure modes and instead incentivizes exploration of the broader reasoning space, which is essential for increasing the probability of discovering a previously inaccessible, correct solution path. This integration transforms the learning process: it not only mitigates diversity collapse in the current task but also aligns with the goals of continual learning. By preserving multiple reasoning modes while anchoring to a correct solution, EVOL-RL avoids forgetting potentially useful strategies and retains knowledge diversity for future tasks. Thus, training under EVOL-RL becomes not only an optimization for present performance but also a proactive investment in future adaptability.

**Reward Formulation.** For each prompt, the policy samples $G$ responses $\{o_i\}_{i=1}^{G}$. Each response is scored on three criteria:

**1. Validity:** The response must provide a numeric final answer in a \boxed{·} format. Responses that fail this check are deemed invalid.

**2. Majority (Selection):** A binary label $y_i \in \{+1, -1\}$ is assigned based on whether a response's answer matches the majority-voted answer from the valid responses. This serves as our selection signal.

**3. Novelty (Variation):** We compute embeddings for the reasoning part of each response to form a cosine similarity matrix. For each response $o_i$, we calculate its mean similarity $\bar{s}_i$ to other responses in the same group (i.e., either majority or minority) and its maximum similarity $m_i$ to any other response in the entire batch. The mean similarity is calculated on an intra-group basis because the majority and minority solutions are often semantically distant; a global mean would be dominated by this gap, obscuring the finer-grained variations among peer solutions within the majority group. The novelty score is:

$$u_i \ = \ 1 - \big(\alpha\,\bar{s}_i + (1 - \alpha)\,m_i\big), \qquad \alpha \in \ (\text{default } 0.5).$$

This score is designed to penalize two distinct forms of redundancy: a high $\bar{s}_i$ indicates conformity to the group's semantic average, while a high $m_i$ flags near-duplication of another specific response.

The score promotes both local and global diversity. Finally, we min-max normalize the scores $\{u_i\}$ separately within the majority and minority groups to get $\tilde{u}_i$. This intra-group normalization is crucial, as it ensures that novelty is measured relative to one's direct peers, allowing for a fair comparison of diversity within each group.

**Final Reward Mapping.** We map the majority label and normalized novelty score into non-overlapping reward bands. This ensures that the selection signal from the majority vote is always prioritized, while novelty refines the reward within each group:

$$
r_i = \begin{cases}
-1, & \text{if invalid;} \\
0.5 + 0.5\,\tilde{u}_i \in [0.5, 1], & \text{if } y_i = +1 \text{ (Majority: higher novelty earns higher reward);} \\
-1 + 0.5\,\tilde{u}_i \in [-1, -0.5], & \text{if } y_i = -1 \text{ (Minority: higher novelty mitigates penalty).}
\end{cases}
$$

Critically, this structure guarantees that any majority solution, regardless of its novelty, receives a higher reward than any minority solution. This maintains a strong pressure towards correctness. More details about the reward implementation are presented in Appendix A.4

**Supporting Mechanisms.** To further reinforce this reward design, we employ two complementary mechanisms. First, within the GRPO objective (Eq. 1), we use an asymmetric clipping range ($\epsilon_{\text{high}} > \epsilon_{\text{low}}$) (Yu et al., 2025). This allows promising and novel solutions with high advantages to receive larger gradient updates, preventing them from being prematurely clipped. Second, we add a token-level entropy regularizer to maintain diversity during the initial generation process:

$$
\mathcal{L}_{\text{ent}}(\theta) = -\lambda_{\text{ent}}\, \mathbb{E}_{o \sim \pi_\theta}\left[\frac{1}{|o|}\sum_{t=1}^{|o|} H(\pi_\theta(\cdot \mid o_{<t}, x))\right], \qquad H(p) = -\sum_v p(v)\log p(v). \tag{2}
$$

The total objective, $\mathcal{L}_{\text{total}} = \mathcal{L}_{\text{GRPO}} + \mathcal{L}_{\text{ent}}$, thus directs learning toward semantically distinct, high-quality responses while maintaining a diverse population of solutions.

### 3.3 HOW EVOL-RL AVOIDS COLLAPSE THROUGH AN EVOLUTIONARY ANALOGY.

EVOL-RL avoids this failure mode by mirroring biological evolution, which balances a stabilizing Selection pressure with a dynamic Variation mechanism. The majority vote acts as our Selection pressure, providing a crucial anchor to correctness. By itself, however, this would lead to a uniform population of solutions, vulnerable to collapse, much like a species with no genetic diversity is vulnerable to a single disease.

To prevent this, our three-part Variation strategy creates a robust exploratory dynamic. The entropy regularizer acts like a higher "mutation rate," ensuring a constant supply of diverse solutions for the system to work with. The novelty reward then provides directional pressure to this variation, giving a "survival bonus" to solutions that are semantically distinct from their peers. In Appendix D, we formally compare the optimal policies induced by a correctness-only objective and by our correctness-plus-novelty objective. With correctness alone, the optimal policy for a given question can be any distribution over the set of correct responses and may even collapse onto a single response. In contrast, under mild assumptions on the similarity structure over responses, our similarity-based novelty reward ensures that any optimal policy spreads probability mass across multiple correct responses, yielding a more diverse solution distribution and stabilizing the policy entropy. Finally, asymmetric clipping ensures that when a highly beneficial "mutation"—a rare, novel, and correct solution—appears, its strong learning signal is fully preserved for the next generation.

This design makes a collapsed state inherently unstable. In a uniform population of near-duplicate solutions, any "mutation" is, by definition, highly novel and receives a higher reward. The learning algorithm is thus forced to shift probability away from the uniform cluster and toward this more promising, distinct solution, ensuring that the policy remains robustly diverse. Also, an analysis of why this reward is appropriate for "almost correct" solutions can also be found in Appendix C.1.

Table 1: Comparison of models trained with TTRL and EVOL-RL. Each cell shows pass@1/pass@16 (averaged on 32 rollouts). Δ uses red (+) for positive and blue for negative values, showing the difference between w/EVOL-RL and w/TTRL.

| Training Dataset | Model | MATH | AIME24 | AIME25 | AMC | GPQA |
|---|---|---|---|---|---|---|
| **Qwen3-4B-Base** | | | | | | |
| – | Base Model | 67.4/89.6 | 10.0/32.4 | 5.5/30.0 | 39.3/75.2 | 34.4/85.6 |
| MATH-TRAIN | w/TTRL | 75.4/86.9 | 12.1/23.2 | 6.8/28.6 | 42.5/75.2 | 36.5/81.4 |
| | w/EVOL-RL | 80.0/93.3 | 20.7/47.6 | 17.5/39.9 | 51.4/80.3 | 37.2/88.7 |
| | Δ | +4.6/+6.4 | +8.6/+24.4 | +10.7/+11.3 | +8.9/+5.1 | +0.7/+7.3 |
| MATH-500 | w/TTRL | 79.3/83.2 | 10.0/28.0 | 7.2/29.9 | 47.6/72.0 | 36.2/75.9 |
| | w/EVOL-RL | 79.8/93.8 | 19.0/43.2 | 16.1/41.9 | 50.3/82.2 | 38.8/89.1 |
| | Δ | +0.5/+10.6 | +9.0/+15.2 | +8.9/+12.0 | +2.7/+10.2 | +2.6/+13.2 |
| AIME24 | w/TTRL | 73.8/84.5 | 16.7/16.7 | 4.6/18.5 | 43.6/65.8 | 35.1/73.5 |
| | w/EVOL-RL | 79.6/93.6 | 20.6/40.9 | 17.1/42.0 | 49.9/80.9 | 38.0/87.8 |
| | Δ | +5.8/+9.1 | +3.9/+24.2 | +12.5/+23.5 | +6.3/+15.1 | +2.9/+14.3 |
| **Qwen3-8B-Base** | | | | | | |
| – | Base Model | 63.6/91.5 | 12.0/39.4 | 8.2/30.8 | 38.7/77.6 | 34.9/88.0 |
| MATH-TRAIN | w/TTRL | 81.1/91.1 | 16.7/37.6 | 15.6/35.9 | 53.6/74.0 | 38.1/77.1 |
| | w/EVOL-RL | 83.6/94.1 | 26.0/51.7 | 21.6/43.1 | 55.5/86.1 | 43.5/88.1 |
| | Δ | +2.5/+3.0 | +9.3/+14.1 | +6.0/+7.2 | +1.9/+12.1 | +5.4/+11.0 |
| MATH-500 | w/TTRL | 85.7/91.9 | 17.7/40.1 | 16.5/34.3 | 51.1/79.1 | 43.5/84.0 |
| | w/EVOL-RL | 84.7/95.1 | 24.1/49.5 | 20.2/44.4 | 58.8/86.0 | 43.9/92.2 |
| | Δ | -1.0/+3.2 | +6.4/+9.4 | +3.7/+10.1 | +7.7/+6.9 | +0.4/+8.2 |
| AIME24 | w/TTRL | 76.8/86.2 | 20.0/20.0 | 11.4/25.4 | 49.5/69.1 | 38.3/74.7 |
| | w/EVOL-RL | 83.1/94.2 | 25.4/38.1 | 16.5/34.7 | 54.4/85.8 | 45.2/90.0 |
| | Δ | +6.3/+8.0 | +5.4/+18.1 | +5.1/+9.3 | +4.9/+16.7 | +6.9/+15.3 |

## 4 EXPERIMENTS

### 4.1 EXPERIMENTAL SETUP

**Benchmarks.** To test our method at scale, we use the large, standard **MATH training set (MATH-TRAIN)** (Hendrycks et al., 2021). We also follow the TTRL (Zuo et al., 2025) by training on two much smaller test sets: the general-purpose **MATH-500** and the competition-level **AIME24** (Li et al., 2024). This comprehensive setup allows us to validate EVOL-RL's versatility across both large-scale and specialized training conditions. Critically, during all training runs, we use only the problem statements, without any ground-truth labels or solutions. For evaluation, we assess the performance of our trained models on a diverse set of five benchmarks to measure both in-domain and out-of-domain generalization. The evaluation suite includes **AIME24**, **AIME25**, **MATH500**, **AMC** (Li et al., 2024), and **GPQA-Diamond (GPQA)** (Rein et al., 2024). Detailed training configuration can be found in Appendix A. Furthermore, we provide additional experimental results on OctoThinker-8B-Hybrid-Base in Appendix B.1. More baseline comparisons, including Entropy Minimization (Agarwal et al., 2025) and Self-Consistency (Wang et al., 2023; Huang et al., 2023), are presented in Appendix B.2. The analysis of the extra time incurred by embedding computation and similarity measurement has been moved to Appendix B.5. A case study in Appendix B.6 shows that embedding similarity reliably captures differences in reasoning paths.

Table 2: Performance of Qwen3-4B-Base with EVOL-RL and its ablations on five benchmarks. Each cell reports pass@1/pass@16 accuracy.

| Training Dataset | Model | MATH | AIME24 | AIME25 | AMC | GPQA |
|---|---|---|---|---|---|---|
| MATH-500 | w/EVOL-RL | **79.8/93.8** | **19.0/43.2** | **16.1/41.9** | **50.3/82.2** | **38.8/89.1** |
| | -ClipHigh | 75.1/91.8 | 12.2/31.8 | 11.4/31.3 | 42.7/73.9 | 32.3/81.8 |
| | -Ent | 79.5/93.4 | 18.3/38.5 | 14.7/34.3 | 48.3/78.6 | 38.6/87.0 |
| | -ClipHigh-Ent | 76.3/92.6 | 12.8/38.8 | 12.5/37.4 | 46.2/77.4 | 35.6/88.8 |
| | -Novelty Reward | 79.3/88.7 | 12.1/27.0 | 11.1/34.8 | 47.6/73.3 | 37.9/81.4 |
| AIME24 | w/EVOL-RL | **79.6/93.6** | **20.6/40.9** | **17.1/42.0** | **49.9/80.9** | **38.0/87.8** |
| | -ClipHigh | 74.1/89.4 | 14.1/26.7 | 8.1/31.1 | 44.6/73.2 | 35.3/81.5 |
| | -Ent | 66.7/89.8 | 10.0/31.4 | 6.6/27.8 | 38.7/74.2 | 34.0/86.2 |
| | -ClipHigh-Ent | 75.3/89.0 | 16.6/26.9 | 9.2/32.2 | 45.8/71.2 | 37.1/82.0 |
| | -Novelty Reward | 79.4/93.0 | 17.7/35.6 | 15.9/37.4 | 48.8/79.6 | 37.9/87.1 |

## 4.2 MAIN RESULTS

The main results of our experiments are summarized in Table 1. We highlight four key findings that demonstrate the superiority of EVOL-RL over the majority-only TTRL baseline.

**EVOL-RL Enhances Both Pass@1 and Pass@16 Performance.** Across all experimental settings, EVOL-RL consistently and substantially improves 'pass@16' performance over TTRL, with gains frequently exceeding 20 percentage points on the most challenging benchmarks (e.g., +24.2% on AIME24 for the 4B model). EVOL-RL also delivers more consistent and substantial improvements to pass@1 accuracy than TTRL. This demonstrates that our method strengthens not only the model's single-shot accuracy but also its ability to explore through multiple attempts.

**Consistent Improvements Across Model Scales and Training Data Sizes.** The benefits of EVOL-RL are robust across both the 4B and 8B model scales, and critically, across training datasets of vastly different sizes. The performance improvements hold true whether training on the large-scale MATH-TRAIN set or the smaller, more specialized MATH-500 and AIME24 sets. This suggests that the underlying mechanism—balancing a majority anchor with novelty-driven rewards—is a fundamental improvement that scales effectively with both model capacity and data volume.

**Strong Cross-Difficulty Robustness** EVOL-RL demonstrates powerful generalization, learning abstract reasoning skills that transfer effectively across different mathematical domains. A powerful example is seen with the 4B model: when trained exclusively on MATH-500, its pass@16 performance on AIME24 and AIME25 is nearly identical to the performance achieved when training on AIME24 directly, confirming that EVOL-RL learns fundamental skills rather than simply overfitting. This effect is further amplified by scale; for the 8B model, training on the large MATH-TRAIN dataset yields pass@1 performance on AIME24 (26.0%) and AIME25 (21.6%) that is far superior to training on AIME24 directly (25.4% and 16.5% respectively). This indicates that EVOL-RL effectively leverages both specialized and large-scale data to build fundamental and transferable reasoning abilities.

**Generalization on Non-Mathematical Reasoning Tasks.** The advantages of EVOL-RL extend beyond the domain of mathematics. On the GPQA benchmark, where TTRL consistently causes pass@16 performance to degrade compared to the base model, EVOL-RL reliably recovers and surpasses the base model. Across all training configurations, it achieves gains of +7 to +15 % in pass@16 over TTRL, indicating that our diversity-preserving reward mechanism fosters a more generalizable reasoning ability that transfers effectively across different domains.

## 4.3 ABLATION STUDY

**Setup.** We conduct an ablation study on EVOL-RL-trained models on Qwen3-4B-Base. EVOL-RL introduces three key modifications compared to the TTRL baseline: (i) the novelty-aware reward function, (ii) a rollout entropy regularizer to encourage exploration, and (iii) an asymmetric PPO

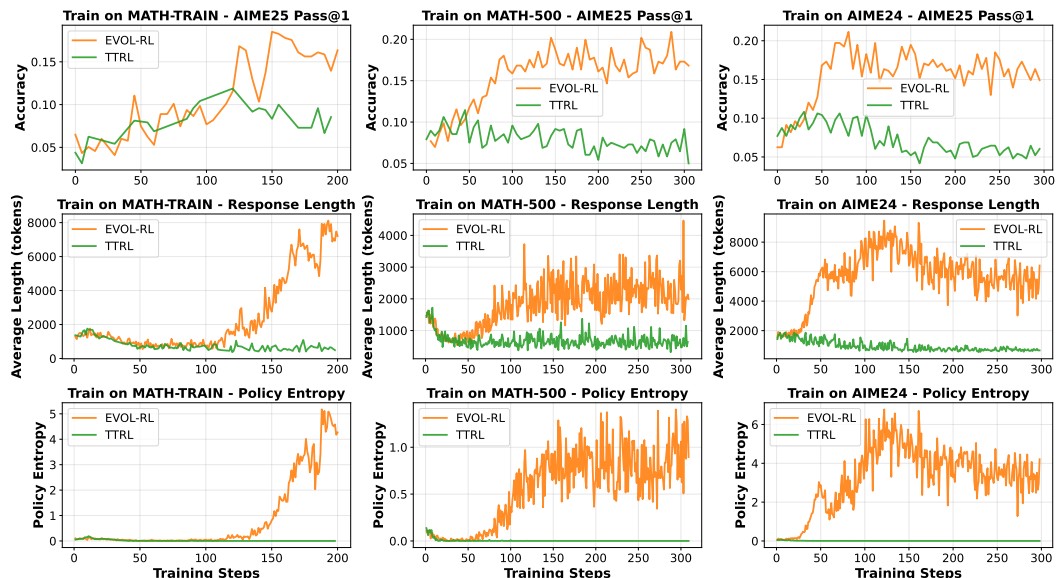

Figure 3: Training dynamics for EVOL-RL and TTRL. **Left:** models trained on *MATH-TRAIN*. **Middle:** models trained on *MATH-500*. **Right:** models trained on *AIME24*. Each panel plots, over training steps, (i) Pass@1 on *AIME25*, (ii) average response length on the training set, and (iii) policy entropy on the training set.

clipping window (higher "ClipHigh") to better preserve learning signals from high-reward samples. We systematically remove these components one at a time ("-Novelty Reward", "-Ent", "-ClipHigh") or in combination. The results are reported in Table 2.

**The Critical Role of Novelty on Easier Datasets.** The importance of the novelty reward is most evident when the model is trained on the MATH-500 dataset. Removing it causes the largest performance degradation in pass@16, especially on the more difficult, out-of-domain AIME24/25. This is because on a dataset with lower complexity, a majority-only approach can quickly cause the model to lock into a single, repetitive reasoning template. Our novelty reward prevents this template lock-in and promotes generalizable skills.

**Exploration Mechanisms as Critical Enablers on Harder Tasks.** On more challenging datasets like AIME24, where the inherent problem difficulty naturally induces a higher baseline of exploration, the other two components become more critical. In this setting, removing the entropy regularizer or the asymmetric clipping consistently lowers pass@16 performance on AIME-style problems. These mechanisms act as crucial enablers for the novelty reward: the entropy regularizer ensures a rich and continuous supply of varied reasoning paths for the novelty selector to act upon, while the higher clipping threshold preserves the full learning signal from rare but high-value solutions.

### 4.4 TRAINING DYNAMICS: HOW EVOL-RL ESCAPES ENTROPY COLLAPSE

To understand the reasons for EVOL-RL's better performance, we analyze its training dynamics in comparison to TTRL in a label-free setting, as shown in Figure 3. An analysis of the training dynamics for the 8B models is presented in Appendix B.4.

**Stage 1: Initial Collapse Under Majority Signal.** Across all three training settings, a consistent initial dynamic unfolds: both EVOL-RL and TTRL show a sharp drop in policy entropy and average response length. This initial phase demonstrates the powerful homogenizing effect of the majority-driven reward, which quickly pushes both models toward short, high-frequency response templates. For TTRL, this collapsed state proves to be permanent; it remains trapped in this low-entropy, low-complexity state for the duration of the training run, regardless of the dataset's scale or difficulty.

**Stage 2: The Evolving Point and Coordinated Recovery.** Following the initial collapse, the training dynamics reveal a crucial divergence centered around a distinct **"evolving point"**. Before this

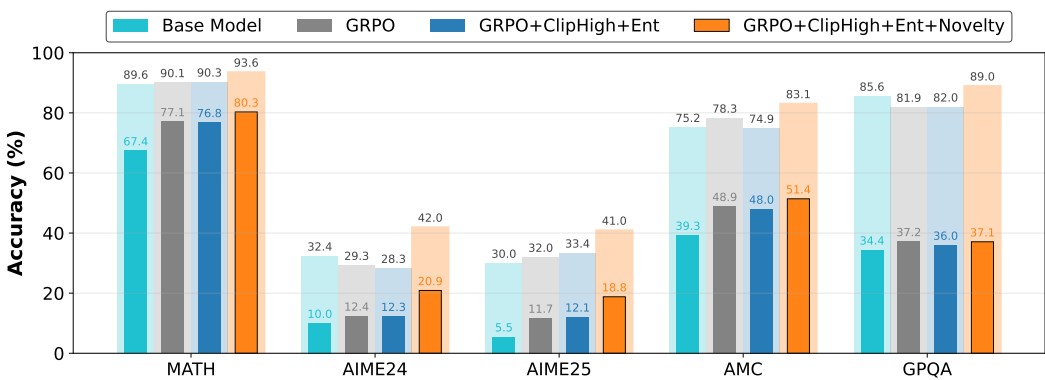

Figure 4: Performance of EVOL-RL's exploration-enhancing components when applied to a standard supervised GRPO baseline. The Qwen3-4B-Base model is trained on the MATH trainig set (Hendrycks et al., 2021) with a ground-truth verifier (RLVR).

Table 3: Generalization performance of the Qwen3-8B-Base model on broader reasoning benchmarks after label-free training on MATH-TRAIN.

| Model | MMLU-Pro | | SuperGPQA | | BBEH | |
|---|---|---|---|---|---|---|
| | Pass@1 | Pass@4 | Pass@1 | Pass@4 | Pass@1 | Pass@4 |
| Qwen3-8B-Base | 47.3 | 74.5 | 26.5 | 54.1 | 10.4 | 24.0 |
| w/TTRL | 53.4 | 73.9 | 29.7 | 53.3 | **12.1** | 24.1 |
| w/EVOL-RL | **55.3** | **78.5** | **30.2** | **57.0** | 11.5 | **24.9** |

point, EVOL-RL's trajectory is nearly indistinguishable from TTRL's; both models exhibit similar performance values and trends, dominated by the majority signal. However, a clear inflection point consistently emerges for EVOL-RL, after which its performance rapidly improves. While the exact timing of this "evolving point" varies across datasets, its appearance is a robust feature of our method. After this "evolving point", EVOL-RL enters a recovery phase characterized by a sustained and coordinated rise across all key metrics: policy entropy breaks away from near-zero values, average response length increases, and out-of-domain accuracy steadily climbs. This coordinated recovery allows the model to reach a new, significantly higher performance plateau where it eventually stabilizes, demonstrating its ability to break free from the majority trap.

EVOL-RL's ability to escape the collapsed state comes from the synergy of its three core components. The entropy regularizer ensures a continuous supply of diverse rollouts, preventing the initial search space from becoming completely uniform. The asymmetric clipping preserves the full gradient signal from the rare but high-value "majority-and-novel" samples that are crucial in the early training phase. Finally, the novelty reward acts as a selection pressure, consistently re-ranking credit within the majority group to favor these distinct solutions over their near-duplicate peers.

### 4.5 EVOL-RL COMPONENTS ALSO STRENGTHEN SUPERVISED GRPO (RLVR)

**Setup.** We apply EVOL-RL's three exploration-enhancing ingredients to a standard supervised GRPO baseline trained on *MATH* training set (Hendrycks et al., 2021) with a ground-truth verifier (RLVR) for two epochs. Figure 4 reports the results.

The primary finding is that the three components are still synergistic, with their full combination yielding the most significant and consistent performance improvements. This complete configuration, `GRPO+ClipHigh+Ent+Novelty`, boosts pass@16 accuracy by 7% to 12% on the challenging out-of-domain AIME24 and AIME25 benchmarks. Crucially, these gains are achieved while also improving pass@1 accuracy, demonstrating that the mechanisms enhance multi-path reliability without sacrificing single-shot performance. This robust improvement extends across all evaluation benchmarks, including the cross-domain GPQA task, demonstrating the great potential of variation reward in a broader context.

### 4.6 GENERALIZATION TO BROADER REASONING BENCHMARKS

To assess whether the reasoning skills enhanced by our method on mathematical data are fundamental and transferable, we evaluate our models on a suite of broader, non-mathematical reasoning benchmarks. After training the Qwen3-8B-Base model on the MATH-TRAIN dataset in a label-free setting, we measure its performance on **MMLU-Pro** (Wang et al., 2024), **SuperGPQA** (Team et al., 2025), and **BBEH** (Kazemi et al., 2025). The results, presented in Table 3, demonstrate that EVOL-RL fosters a more generalizable reasoning ability compared to TTRL.

A contrasting pattern emerges between the two methods. While TTRL shows clear improvements over the base model on pass@1 accuracy, its effect on pass@4 is less consistent, falling slightly below the base model's performance on SuperGPQA and BBEH. This pattern is consistent with our findings on the mathematical reasoning tasks, where the narrow focus of the consensus-only objective can hurt multi-path reliability. In contrast, EVOL-RL demonstrates a more robustly positive transfer of skills, improving upon both the base model and TTRL across pass@1 and pass@4 metrics. For example, on MMLU-Pro, EVOL-RL achieves a pass@4 score of 78.5%, a clear improvement over TTRL's 73.9%. This indicates that our principle of encouraging diverse reasoning helps the model learn more fundamental skills that generalize effectively beyond mathematics.

## 5 CONCLUSION

In this work, we diagnose the entropy collapse, a critical failure mode in LLM evolving where majority-only rewards suppress solution diversity and harm generalization. To solve this, we propose EVOL-RL, a framework that balances the stability of majority-vote selection with an explicit variation incentive that rewards semantic novelty. Our experiments demonstrate that EVOL-RL successfully prevents collapse by maintaining policy entropy and reasoning complexity, which translates into substantial performance gains on both in-domain and out-of-domain benchmarks. By anchoring learning to a stable majority signal while simultaneously encouraging exploration, EVOL-RL offers a robust and practical methodology for enabling LLMs to continuously and autonomously evolve without external labels.

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

# A IMPLEMENTATION DETAILS

This section provides additional details on the implementation of our reward formulation and supporting mechanisms.

## A.1 TRAINING CONFIGURATION.

We conduct our experiments on two recent open-source base models: **Qwen3-4B-Base** and **Qwen3-8B-Base**. Our training process is implemented using the GRPO algorithm. We adopt a setup similar to that of TTRL for generating training signals. For each problem instance, we first perform a rollout phase where the policy generates 64 candidate responses. A majority label is then determined by performing a majority vote on the final answers extracted from these 64 samples. Subsequently, a random subset of 32 of these responses is used to form a batch for a single model update step. To ensure that the model has sufficient capacity for complex, multi-step reasoning, we set the maximum response length to 12,288 tokens during generation. To guide the model's reasoning process, we utilize the system prompt from SimpleRL-Zoo (Zeng et al., 2025). Implementation details are discussed in Appendix A.

## A.2 SYSTEM PROMPT

For all experiments, we used the following system prompt to guide the model's generation format, ensuring that it produces a step-by-step reasoning process and a clearly marked final answer (Zeng et al., 2025):

> **System Prompt**
>
> ```
> Please reason step by step, and put your final answer within
> \boxed{}.
> ```

## A.3 ANSWER AND REASONING EXTRACTION

To implement the scoring criteria described in the main text, we apply the following extraction procedure for each generated response $o_i$:

- **Final Answer Extraction (for Validity):** We parse the response to find the content within the final occurrence of the $\boxed{\cdot}$ command. A response is deemed "valid" only if this command is present and its content contains at least one numeric digit. This extracted numeric string is used for the majority vote.

## A.4 NOVELTY SCORE CALCULATION DETAILS

The novelty score $u_i$ relies on computing semantic similarity between the reasoning parts of the generated responses.

**Embedding Model.** We use the **Qwen3-4B-Embedding** model to generate dense vector representations for the extracted reasoning parts. Each vector is L2-normalized before similarity computation.

**Cosine Similarity Matrix.** For a group of $G$ responses with corresponding L2-normalized embedding vectors $\{\mathbf{v}_1, \ldots, \mathbf{v}_G\}$, the cosine similarity matrix $\mathbf{S} \in \mathbb{R}^{G \times G}$ is computed as $\mathbf{S} = \mathbf{V}\mathbf{V}^T$, where $\mathbf{V}$ is the matrix whose rows are the vectors $\mathbf{v}_i$. The element $S_{ij}$ represents the cosine similarity between the reasoning of response $o_i$ and $o_j$.

**Intra-Group Min-Max Normalization.** To obtain the normalized novelty score $\tilde{u}_i \in [0, 1]$ from the raw scores $\{u_k\}$ within a specific group (e.g., the majority group), we apply standard min-max normalization:

$$\tilde{u}_i = \frac{u_i - \min(\{u_k\})}{\max(\{u_k\}) - \min(\{u_k\}) + \epsilon_{\text{norm}}}$$

where $\epsilon_{\text{norm}}$ is a small constant (e.g., $10^{-8}$) to prevent division by zero in cases where all novelty scores in the group are identical.

## A.5 HYPERPARAMETER SETTINGS

For our label-free experiments, we largely follow the settings established by TTRL to ensure a fair comparison. The general hyperparameters are detailed in Table 4, and the settings specific to our EVOL-RL method are listed in Table 5.

Table 4: General hyperparameters for label-free training, following TTRL.

| Hyperparameter | Value |
|---|---|
| Train Batch Size | 8 |
| PPO Mini-Batch Size | 1 (effective size of 32) |
| PPO Micro-Batch Size | 2 |
| Rollouts for Majority Vote | 64 |
| Rollouts Used for Training | 32 |
| Generation Temperature | 1.0 |
| Validation Temperature | 0.6 |
| Learning Rate | 5e-7 |
| Use KL Loss | True |
| KL Loss Coefficient | 0.001 |

Table 5: Key hyperparameters specific to the EVOL-RL framework.

| Hyperparameter | Value |
|---|---|
| Asymmetric Clipping High ($\epsilon_{\text{high}}$) | 0.28 |
| Entropy Regularizer Coefficient ($\lambda_{\text{ent}}$) | 0.003 |
| Novelty Score Mixing Coefficient ($\alpha$) | 0.5 |

## A.6 COMPUTATIONAL RESOURCES

All experiments reported in this paper were conducted on a single server equipped with 8x NVIDIA H20 GPUs.

## B ADDITIONAL EXPERIMENTAL RESULTS

### B.1 EFFECTIVENESS ON DIFFERENT MODEL ARCHITECTURES

To verify that our approach is not limited to a single model family, we conducted an additional experiment applying EVOL-RL and TTRL to the **OctoThinker-8B-Hybrid-Base** model (Wang et al., 2025c), a different architecture from the Qwen3 series. We used the same label-free training setup on the MATH-500 dataset.

The results, presented in Table 6, strongly confirm our core thesis. The TTRL baseline, when applied to OctoThinker, exhibits the classic symptoms of entropy collapse: while it significantly improves in-domain `pass@1` accuracy (e.g., on MATH, from 33.8% to 63.8%), it fails to improve multi-path accuracy. In fact, `pass@16` performance degrades on AIME24 (from 13.5% to 10.8%) and GPQA (from 85.7% to 71.5%) compared to the base model.

In sharp contrast, EVOL-RL successfully prevents this collapse and translates exploration into robust performance gains. While achieving a comparable `pass@1` improvement on MATH, EVOL-RL yields massive improvements in `pass@16` across all benchmarks. Most notably, it achieves a +19.5% gain on AIME24 and a +11.4% gain on AIME25 in `pass@16` accuracy over TTRL.

Table 6: Comparison of models trained with TTRL and EVOL-RL on MATH-500 on OctoThinker-8B-Hybrid-Base. Each cell shows pass@1/pass@16 (averaged over 32 rollouts). Δ uses red (+) for positive and blue for negative values.

| Model | MATH | AIME24 | AIME25 | AMC | GPQA |
|---|---|---|---|---|---|
| **OctoThinker-8B-Hybrid-Base** | | | | | |
| Base Model | 33.8/79.8 | 1.5/13.5 | 1.3/13.9 | 16.2/56.9 | 26.3/85.7 |
| w/TTRL | 63.8/76.4 | 2.8/10.8 | 2.1/11.0 | 27.9/54.7 | 31.9/71.5 |
| w/EVOL-RL | 63.2/86.3 | 9.0/30.3 | 7.2/22.4 | 34.1/65.6 | 33.2/85.7 |
| Δ | -0.6/+9.9 | +6.2/+19.5 | +5.1/+11.4 | +6.2/+10.9 | +1.3/+14.2 |

Table 7: Extended comparison of baseline methods on Qwen3-4B-Base, trained in a label-free setting using MATH-Train. Each cell shows pass@1/pass@16, and the highest value in each column is bolded.

| Model | MATH | AIME24 | AIME25 | AMC | GPQA |
|---|---|---|---|---|---|
| **Qwen3-4B-Base** | | | | | |
| Base Model | 67.4/89.6 | 10.0/32.4 | 5.5/30.0 | 39.3/75.2 | 34.4/85.6 |
| w/TTRL | 75.4/86.9 | 12.1/23.2 | 6.8/28.6 | 42.5/75.2 | 36.5/81.4 |
| w/EM-RL-Token | 76.0/90.6 | 12.5/31.3 | 10.5/30.8 | 46.6/77.7 | 36.8/82.6 |
| w/EM-RL-Sequence | 67.4/89.9 | 10.6/31.4 | 7.1/28.8 | 39.6/73.7 | 34.5/86.2 |
| w/Self-Consistency | 76.0/89.7 | 12.5/30.4 | 10.4/33.6 | 48.1/78.1 | 35.9/81.2 |
| **w/EVOL-RL** | **80.0/93.3** | **20.7/47.6** | **17.5/39.9** | **51.4/80.3** | **37.2/88.7** |

This experiment demonstrates that entropy collapse is a fundamental flaw of the majority-only objective and that EVOL-RL is a robust and generalizable solution that functions effectively across different model architectures.

## B.2    COMPARISON WITH OTHER SELF-IMPROVEMENT BASELINES

To demonstrate that EVOL-RL is not just an improvement over TTRL but a more robust solution to the "entropy collapse" problem, we compare it against a broader suite of label-free self-improvement methods. These include methods based on self-consistency (Self-Consistency (Wang et al., 2023; Huang et al., 2023)) and intrinsic confidence (EM-RL-Token and EM-RL-Sequence (Agarwal et al., 2025)). We trained all methods on the Qwen3-4B-Base model under the same label-free setting, with results presented in Table 7.

The results highlight a clear and consistent pattern: methods that optimize for a single signal (like consensus or confidence) fail to achieve robust, generalizable gains.

While baselines like TTRL, EM-RL-Token, and Self-Consistency all show moderate improvements in `pass@1` accuracy on some benchmarks, they don't show any consistent improvement in `pass@16` performance, which is an indicator of entropy collapse. On the challenging AIME24 benchmark, every single one of these baselines performs worse than the original Base Model on `pass@16` (e.g., 23.2% for TTRL and 30.4% for Self-Consistency, vs. 32.4% for the Base Model). This strongly suggests that their singular focus on "certainty" actively degrades solution diversity and multi-path reasoning.

In stark contrast, EVOL-RL is the only method that robustly improves both single-shot accuracy (`pass@1`) and multi-path reliability (`pass@16`) across all benchmarks. The gains are most pronounced on out-of-domain tasks. On AIME24, EVOL-RL achieves a `pass@1` of **20.7%** (vs. ~12.5% for the next-best baselines) and a `pass@16` of **47.6%**, demonstrating a massive +15% improvement over even the Base Model, whereas all other methods failed. This result strongly supports our central thesis: a simple consensus or confidence signal is insufficient. True self-improvement

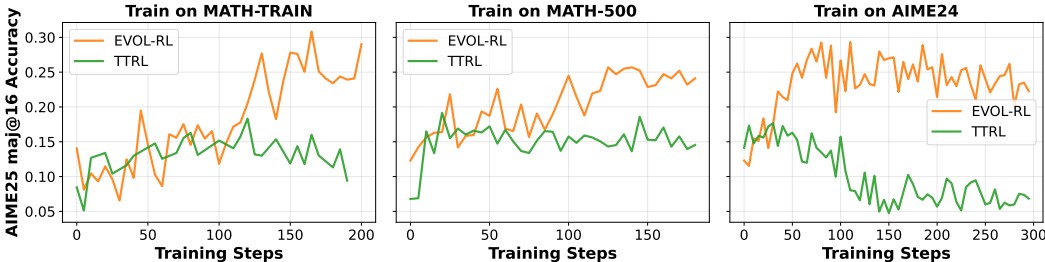

Figure 5: Training dynamics of the majority-vote accuracy (maj@16) for EVOL-RL and TTRL. Each panel plots the accuracy of the consensus answer derived from 16 rollouts over the course of training. The training datasets are: (**Left**) MATH-TRAIN, (**Middle**) MATH-500, and (**Right**) AIME24.

requires an explicit mechanism—like our *majority-for-stability + novelty-for-exploration* rule—to prevent entropy collapse and foster diverse, generalizable reasoning.

### B.3 ANALYSIS OF THE MAJORITY VOTE SIGNAL

To further investigate the differences between EVOL-RL and TTRL, we analyze the quality of the training signal itself by tracking the accuracy of the majority vote (maj@16) over the course of training, as shown in Figure 5. This analysis reveals how the self-generated pseudo-labels evolve under each method.

A highly consistent pattern emerges across all three training datasets. TTRL initially improves the maj@16 accuracy over the base model, but it quickly converges to a performance plateau. For the remainder of the training, its maj@16 accuracy remains largely unchanged, indicating that the consensus-only approach rapidly finds a local optimum for the consensus answer and becomes locked in, unable to discover better solutions.

In contrast, EVOL-RL exhibits a markedly different dynamic. While its initial trajectory often mirrors that of TTRL, reflecting the early stabilizing influence of the consensus signal, a clear divergence occurs. Consistent with the inflection point observed in our main training dynamics analysis, EVOL-RL's maj@16 accuracy breaks away from the TTRL plateau and begins a second, sustained ascent. It reliably climbs to and stabilizes at a significantly higher level of accuracy. This demonstrates that EVOL-RL's exploration mechanisms not only improve the final policy but also progressively refine the quality of the pseudo-labels used for training, allowing the model to escape suboptimal consensus and continuously improve its understanding of the task.

### B.4 TRAINING DYNAMICS OF 8B MODELS

The training dynamics of the 8B models, presented in Figure 6, largely mirror the patterns observed with the 4B models, confirming that the core mechanisms of EVOL-RL are robust to scale.

Across all three training datasets (MATH-TRAIN, MATH-500, and AIME24), we observe the same two-stage process. In Stage 1, both TTRL and EVOL-RL experience an initial drop in policy entropy and response length due to the strong initial pressure of the majority-vote signal. TTRL becomes permanently trapped in this low-entropy, low-complexity state.

In Stage 2, EVOL-RL consistently diverges at an "evolving point." Its policy entropy begins a sustained recovery, followed by a coordinated increase in average response length and out-of-domain accuracy on AIME25. This confirms that even at a larger scale, EVOL-RL successfully prevents entropy collapse and fosters a positive feedback loop where exploration, reasoning complexity, and performance reinforce one another, while the consensus-only TTRL approach stagnates.

### B.5 ANALYSIS OF THE COMPUTATIONAL OVERHEAD FROM THE NOVELTY REWARD

A practical consideration for our method is the additional computational load introduced by the embedding-based novelty reward. To rigorously quantify this, we tracked the wall-clock time per

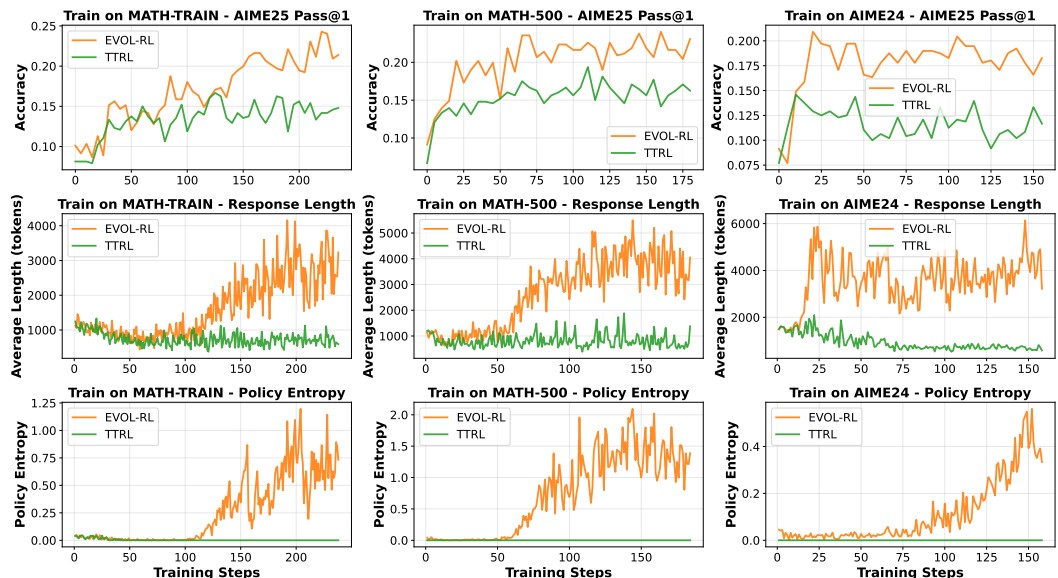

Figure 6: Training dynamics for EVOL-RL and TTRL on Qwen3-8B-Base model. **Left:** models trained on *MATH-TRAIN*. **Middle:** models trained on *MATH-500*. **Right:** models trained on *AIME24*. Each panel plots, over training steps, (i) Pass@1 on *AIME25*, (ii) average response length on the training set, and (iii) policy entropy on the training set.

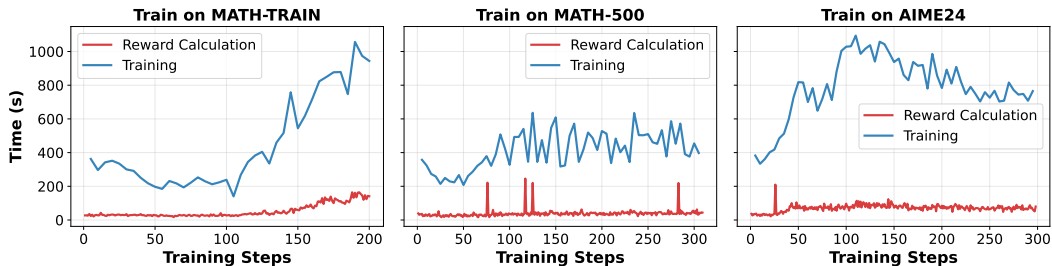

Figure 7: Wall-clock time (seconds) per training step for **Novelty Reward Calculation** (Red) versus the **Total Training Process** (Blue) across three datasets. While both costs naturally increase as the model evolves to generate longer reasoning paths, the reward calculation overhead remains a minor fraction of the total computational load.

training step, decomposing it into two parts: (1) the Novelty Reward Calculation (comprising $B \times N$ embedding model calls and the $O(BN^2)$ similarity matrix computation, where $B$ is the batch size and $N$ is the number of rollouts), and (2) the Total Training Process (including rollout generation, reward calculation and model updates). The results are plotted in Figure 7.

The analysis reveals two key insights:

- **Scaling with Reasoning Length:** As expected, both curves rise over time. This correlates with our findings in Figure 3 that the model learns to generate significantly longer CoT during training. Longer responses require more time to generate (Training) and more time to embed (Reward Calculation).

- **Low Relative Overhead:** while the reward cost scales with response length (requiring embedding of longer sequences), the Novelty Reward Calculation (red curve) remains a small portion of the overall runtime, stabilizing around 100 seconds.

The occasional sharp spikes in the Red curve are anomalous and likely attributable to external API connection instabilities. We conclude that the embedding-based reward introduces a manageable

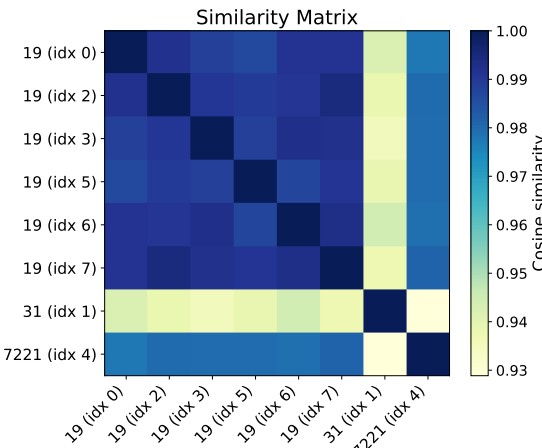

Figure 8: Case study of the reasoning path cosine similarity matrix for 8 rollouts on a single AIME25 problem. The axes are labeled with the `final_answer(index_i)` for each rollout. The 8 rollouts produced three distinct final answers: "19" (6 times), "31" (1 time), and "7221" (1 time).

overhead that is marginal compared to the inherent cost of training reasoning models, especially given the substantial gains in performance and generalization it unlocks.

### B.6 CASE STUDY: VALIDITY OF THE SEMANTIC NOVELTY PROXY

A potential concern regarding our method is whether semantic similarity in an embedding space is a meaningful proxy for genuine reasoning novelty. To validate this, we conducted a case study, shown in Figure 8, by generating 8 rollouts for a single AIME25 problem using Qwen3-4B-Instruct-2407 and computing their reasoning path similarity matrix with Qwen3-4B-Embedding.

The results provide strong empirical evidence for the validity of this proxy. As shown in the figure, the matrix reveals distinct, high-similarity clusters that align with the reasoning logic. Specifically, the 6 rollouts that produced the same final answer ("19") have an extremely high intra-cluster similarity (the 6x6 block at the top-left). Crucially, these paths are semantically distinct from the path that answered "31" and the path that answered "7221", showing clear separation in the embedding space.

This demonstrates that the embedding space does successfully capture and cluster the core logic of the reasoning paths, distinguishing between different reasoning trajectories.

## C ADDITIONAL RATIONALE SUPPORTING THE REWARD DESIGN

### C.1 ANALYSIS OF REWARD FORMULATION ON "ALMOST CORRECT" SOLUTIONS

A key nuance in our reward formulation is how it handles solutions that are "almost correct"— for instance, a minority solution that is semantically similar to a correct majority path but fails on a minor step. This section provides a detailed analysis of how our two-part novelty score, $u_i = 1 - (\alpha \bar{s}_i + (1 - \alpha)m_i)$, is specifically designed to handle this nuance.

Our primary goal is to apply strong negative pressure against high-frequency, common failure modes, not to indiscriminately punish all incorrect explorations. In the scenario where a minority solution is semantically similar to the majority (resulting in a high global similarity $m_i$), the intra-group mean similarity $\bar{s}_i$ (the solution's average similarity to other minority solutions) becomes the critical differentiator. We analyze two distinct cases:

- **Case 1: The error is a rare, exploratory mistake.** If the "almost correct" error is a rare, occasional mistake, the reasoning path will be semantically dissimilar from the other failure modes in the minority group. This results in a low $\bar{s}_i$. According to our novelty

formula, this low $\bar{s}_i$ counteracts the high $m_i$, leading to a higher overall novelty score $u_i$. This effectively mitigates the penalty, ensuring the model is not strongly discouraged from valid exploration.

- **Case 2: The error is a common failure mode.** If the error represents a high-frequency failure mode (e.g., a consistent arithmetic error), the reasoning path will be semantically similar to many other solutions in the minority group. This results in a high $\bar{s}_i$. In this case, both the $\bar{s}_i$ and $m_i$ terms are high, leading to a very low $u_i$. This results in a maximum penalty, preventing the model from collapsing into a "consistent but wrong" state.

This design ensures that our reward mechanism is robust. It relies on the group-level context to selectively protect valuable explorations while aggressively pruning systematic errors. This is particularly crucial in high-uncertainty scenarios where the novelty signal must accurately guide exploration, which is the exact behavior we aim to encourage.

## D  THEORETICAL JUSTIFICATION

In this section, we formally justify the different behaviors of the optimal policy under correctness-only reinforcement learning and under our similarity-augmented objective. We show that, for a suitably small similarity weight and under a simple similarity-gap structure, both objectives have globally optimal policies that concentrate on the set of correct reasoning traces, but our similarity-augmented objective further selects solutions with strictly more diverse coverage over correct trajectories and, under additional symmetry condition, maximal policy entropy on the correct set.

### D.1  SETUP

For a given reasoning question $q$, there is a finite set $\mathcal{Y}$ of complete chain-of-thought (CoT) trajectories for $q$. Each trajectory receives a binary task reward

$$r(y) \in \{0, 1\}, \qquad y \in \mathcal{Y}.$$

We denote the correct set (answer-correct set) by

$$G := \{y \in \mathcal{Y} : r(y) = 1\},$$

and assume $G \neq \emptyset$. A policy $\pi$ maps question to chain of thought reasoning. For simplicity, we will omit the condition of question in the following analysis.

**Objectives.**  The correctness-only objective is

$$J_0(\pi) := \mathbb{E}_\pi[r(Y)] = \sum_{y \in \mathcal{Y}} \pi(y)\, r(y).$$

To encourage diverse correct solutions, we introduce a symmetric, nonnegative CoT-level dissimilarity

$$d : \mathcal{Y} \times \mathcal{Y} \to [0, \infty), \quad d(y, y') = d(y', y), \quad d(y, y) = 0,$$

and assume that $d$ is bounded:

$$D_{\max} := \max_{y, y' \in \mathcal{Y}} d(y, y') < \infty. \tag{3}$$

Given a group size $K \geq 2$ and similarity weight $\lambda > 0$, we form i.i.d. samples $Y_1, \ldots, Y_K \sim \pi$ and define the group reward

$$R_{\text{group}}(Y_{1:K}) := \frac{1}{K} \sum_{k=1}^{K} r(Y_k) + \lambda \cdot \frac{1}{K(K-1)} \sum_{k \neq \ell} d(Y_k, Y_\ell).$$

The correctness together with similarity objective is the expected group reward

$$J_\lambda(\pi) := \mathbb{E}_{Y_{1:K} \sim \pi}[R_{\text{group}}(Y_{1:K})].$$

The following standard computation shows that $J_\lambda$ is a quadratic functional of $\pi$.

**Lemma 1** (Expected group reward as a quadratic in $\pi$). *For any $\pi \in \Delta(\mathcal{Y})$,*

$$J_\lambda(\pi) = \sum_y \pi(y)\, r(y) + \lambda \sum_{y,y'} \pi(y)\pi(y')\, d(y,y'). \tag{4}$$

*Proof.* By linearity of expectation and the i.i.d. assumption on $Y_{1:K}$, the first term yields

$$\mathbb{E}\Big[\frac{1}{K}\sum_{k=1}^{K} r(Y_k)\Big] = \mathbb{E}[r(Y_1)] = \sum_y \pi(y)r(y).$$

For the second term, each ordered pair $(k,\ell)$ with $k \neq \ell$ has the same distribution, so

$$\mathbb{E}\Big[\frac{1}{K(K-1)}\sum_{k\neq\ell} d(Y_k, Y_\ell)\Big] = \mathbb{E}[d(Y_1, Y_2)] = \sum_{y,y'} \pi(y)\pi(y')d(y,y'),$$

which gives equation 4. $\qquad\square$

## D.2 STEP 1: GLOBAL MAXIMIZERS CONCENTRATE ON THE CORRECT SET

We first show that for a suitable range of $\lambda$, any global maximizer of $J_0$ or $J_\lambda$ puts zero probability on incorrect trajectories.

**Lemma 2** (Global maximizers of $J_0$ are supported on $G$). *Assume $G \neq \emptyset$. Then any global maximizer $\pi^\star$ of $J_0$ satisfies*

$$\mathrm{supp}(\pi^\star) \subseteq G.$$

*Moreover, the set of global maximizers of $J_0$ is exactly $\Delta(G)$.*

*Proof.* For any $\pi \in \Delta(\mathcal{Y})$,

$$J_0(\pi) = \sum_y \pi(y)r(y) = \sum_{y \in G} \pi(y) = \pi(G) \leq 1,$$

with equality iff $\pi(G) = 1$, i.e., $\mathrm{supp}(\pi) \subseteq G$. Because $G \neq \emptyset$, there exists a policy with $\pi(G) = 1$ and $J_0(\pi) = 1$, so the maximal value of $J_0$ is 1, achieved exactly by policies supported on $G$. $\qquad\square$

For the similarity-augmented objective, the task reward gap between correct and incorrect trajectories is 1, while the similarity term is bounded by $D_{\max}$ from equation 3. For sufficiently small $\lambda$, the former always dominates the latter.

**Lemma 3** (For small $\lambda$, global maximizers of $J_\lambda$ are supported on $G$). *Suppose $G \neq \emptyset$ and $D_{\max}$ is defined as in equation 3. If*

$$0 < \lambda < \frac{1}{2D_{\max}}, \tag{5}$$

*(with the convention that the condition is vacuous if $D_{\max} = 0$), then any global maximizer $\pi^\star$ of $J_\lambda$ satisfies*

$$\mathrm{supp}(\pi^\star) \subseteq G.$$

*Proof.* Using equation 4, we can write

$$J_\lambda(\pi) = \sum_y \pi(y)r(y) + \lambda \sum_{y,y'} \pi(y)\pi(y')d(y,y').$$

For $y \in \mathcal{Y}$, the partial derivative of $J_\lambda$ w.r.t. $\pi(y)$ is

$$g_y(\pi) := \frac{\partial J_\lambda(\pi)}{\partial \pi(y)} = r(y) + 2\lambda \sum_{y'} \pi(y')d(y,y'),$$

where we used the symmetry $d(y,y') = d(y',y)$.

Assume for contradiction that $\pi^\star$ is a global maximizer of $J_\lambda$ and there exists an incorrect trajectory $y_0 \notin G$ with $\pi^\star(y_0) > 0$. Because $G \neq \emptyset$, pick any $y_1 \in G$ (so $r(y_1) = 1$). Consider the feasible direction in the simplex

$$v := e_{y_1} - e_{y_0},$$

which corresponds to infinitesimally moving probability mass from $y_0$ to $y_1$. For sufficiently small $\varepsilon > 0$, the perturbed policy $\pi^\star + \varepsilon v$ remains in $\Delta(\mathcal{Y})$.

The directional derivative of $J_\lambda$ at $\pi^\star$ along $v$ is

$$\frac{d}{d\varepsilon} J_\lambda(\pi^\star + \varepsilon v) \Big|_{\varepsilon=0} = \langle \nabla J_\lambda(\pi^\star), v \rangle = g_{y_1}(\pi^\star) - g_{y_0}(\pi^\star).$$

By definition of $g_y$,

$$g_{y_1}(\pi^\star) - g_{y_0}(\pi^\star) = \big(r(y_1) - r(y_0)\big) + 2\lambda \sum_{y'} \pi^\star(y')\big(d(y_1, y') - d(y_0, y')\big)$$

$$= 1 + 2\lambda \sum_{y'} \pi^\star(y')\big(d(y_1, y') - d(y_0, y')\big),$$

since $r(y_1) = 1$ and $r(y_0) = 0$. By the definition of $D_{\max}$,

$$\big| d(y_1, y') - d(y_0, y') \big| \leq D_{\max} \quad \text{for all } y',$$

and since $\sum_{y'} \pi^\star(y') = 1$, we obtain

$$\left| \sum_{y'} \pi^\star(y')\big(d(y_1, y') - d(y_0, y')\big) \right| \leq D_{\max}.$$

Hence

$$g_{y_1}(\pi^\star) - g_{y_0}(\pi^\star) \geq 1 - 2\lambda D_{\max}.$$

If $0 < \lambda < 1/(2D_{\max})$, then $1 - 2\lambda D_{\max} > 0$, so

$$\langle \nabla J_\lambda(\pi^\star), v \rangle > 0.$$

Thus moving a small amount of mass from $y_0$ to $y_1$ strictly increases $J_\lambda$, contradicting the assumption that $\pi^\star$ is a (global, hence local) maximizer. Therefore no maximizer can assign positive probability to any incorrect trajectory, and $\mathrm{supp}(\pi^\star) \subseteq G$. $\qquad\square$

Combining Lemma 2 and Lemma 3, we obtain that assume $G \neq \emptyset$, $D_{\max} < \infty$, and equation 5. Then any global maximizer of $J_0$ or $J_\lambda$ is supported on $G$. Thus we may restrict attention to

$$\Delta(G) := \{\pi \in \Delta(\mathcal{Y}) : \mathrm{supp}(\pi) \subseteq G\}$$

when comparing optimal policies.

### D.3 STEP 2: MODE STRUCTURE INSIDE $G$ AND A PIECEWISE-HOMOGENEOUS SIMILARITY GAP

We now formalize the assumption that the correct set $G$ contains multiple qualitatively distinct solution modes, with large similarity gaps between modes and relatively homogeneous dissimilarity within each mode.

**Assumption 1** (Mode partition of correct CoTs). *The correct set $G$ is partitioned into $M \geq 2$ disjoint subsets*

$$G = G_1 \cup \cdots \cup G_M, \qquad G_m \cap G_{m'} = \emptyset \ (m \neq m'),$$

*where each $G_m$ corresponds to a distinct reasoning mode (solution pattern). Let $N_m := |G_m|$ and $N := |G| = \sum_{m=1}^M N_m$.*

For any $\pi \in \Delta(G)$, we define the total probability mass on each mode:

$$w_m(\pi) := \sum_{y \in G_m} \pi(y), \qquad m = 1, \ldots, M.$$

Then $w(\pi) = (w_1(\pi), \ldots, w_M(\pi))$ lies in the simplex

$$\Delta_M := \{w \in \mathbb{R}_+^M : \sum_{m=1}^M w_m = 1\}.$$

We impose a piecewise-homogeneous similarity-gap assumption on $d$ within $G$.

**Assumption 2** (Piecewise-homogeneous dissimilarity on the correct set). *There exist constants $D_{\text{in}} \geq 0$ and $D_{\text{out}} > D_{\text{in}}$ such that for all $y, y' \in G$,*

$$d(y, y') = \begin{cases} 0, & \text{if } y = y', \\ D_{\text{in}}, & \text{if } y \neq y', \ y, y' \in G_m \text{ for some } m \text{ (same mode)}, \\ D_{\text{out}}, & \text{if } y \in G_m, \ y' \in G_{m'} \ (m \neq m') \text{ (different modes)}. \end{cases}$$

Assumption 2 idealizes the intuitive condition that trajectories within the same mode are relatively similar (with dissimilarity $D_{\text{in}}$), while trajectories from different modes are more dissimilar (with $D_{\text{out}} > D_{\text{in}}$).

Under these assumptions, the similarity term in $J_\lambda$ over $\Delta(G)$ admits a convenient decomposition into a mode-level and an intra-mode component.

**Lemma 4** (Decomposition of the dissimilarity term on $G$). *Under Assumptions 1 and 2, for any $\pi \in \Delta(G)$,*

$$\sum_{y, y' \in G} \pi(y)\pi(y')d(y, y') = D_{\text{out}} + (D_{\text{in}} - D_{\text{out}}) \sum_{m=1}^M w_m(\pi)^2 - D_{\text{in}} \sum_{y \in G} \pi(y)^2.$$

*Consequently, on $\Delta(G)$,*

$$J_\lambda(\pi) = 1 + \lambda\Big[D_{\text{out}} + (D_{\text{in}} - D_{\text{out}}) \sum_{m=1}^M w_m(\pi)^2 - D_{\text{in}} \sum_{y \in G} \pi(y)^2\Big]. \tag{6}$$

*Proof.* Because $\text{supp}(\pi) \subseteq G$, we have $r(y) = 1$ on $G$ and $r(y) = 0$ otherwise, hence $\sum_y \pi(y)r(y) = \sum_{y \in G} \pi(y) = 1$, giving the first term in equation 6.

For the dissimilarity term, we decompose by modes:

$$\sum_{y, y' \in G} \pi(y)\pi(y')d(y, y') = \sum_{m=1}^M \sum_{y, y' \in G_m} \pi(y)\pi(y')d(y, y') + \sum_{m \neq m'} \sum_{y \in G_m, y' \in G_{m'}} \pi(y)\pi(y')d(y, y').$$

Within a fixed mode $G_m$, we have $d(y, y) = 0$ and $d(y, y') = D_{\text{in}}$ for $y \neq y'$, so

$$\sum_{y, y' \in G_m} \pi(y)\pi(y')d(y, y') = D_{\text{in}} \sum_{\substack{y, y' \in G_m \\ y \neq y'}} \pi(y)\pi(y')$$

$$= D_{\text{in}}\Big[\Big(\sum_{y \in G_m} \pi(y)\Big)^2 - \sum_{y \in G_m} \pi(y)^2\Big]$$

$$= D_{\text{in}}\big[w_m(\pi)^2 - \sum_{y \in G_m} \pi(y)^2\big].$$

Summing over $m$ yields

$$\sum_{m=1}^M \sum_{y, y' \in G_m} \pi(y)\pi(y')d(y, y') = D_{\text{in}} \sum_{m=1}^M w_m(\pi)^2 - D_{\text{in}} \sum_{y \in G} \pi(y)^2.$$

Across different modes, $d(y, y') = D_{\text{out}}$ whenever $y \in G_m$ and $y' \in G_{m'}$ with $m \neq m'$, hence

$$\sum_{m \neq m'} \sum_{y \in G_m, y' \in G_{m'}} \pi(y)\pi(y')d(y, y') = D_{\text{out}} \sum_{m \neq m'} w_m(\pi)w_{m'}(\pi).$$

Since $\sum_m w_m(\pi) = 1$,

$$\sum_{m \neq m'} w_m w_{m'} = \left(\sum_m w_m\right)^2 - \sum_m w_m^2 = 1 - \sum_m w_m^2.$$

Combining the within-mode and cross-mode contributions gives

$$\sum_{y, y' \in G} \pi(y)\pi(y')d(y, y') = D_{\text{in}} \sum_m w_m^2 - D_{\text{in}} \sum_{y \in G} \pi(y)^2 + D_{\text{out}}\left(1 - \sum_m w_m^2\right)$$

$$= D_{\text{out}} + (D_{\text{in}} - D_{\text{out}}) \sum_m w_m^2 - D_{\text{in}} \sum_{y \in G} \pi(y)^2.$$

Substituting this into Lemma 1 with $\sum_y \pi(y)r(y) = 1$ yields equation 6. $\qquad\square$

Thus, over the feasible region $\Delta(G)$, maximizing $J_\lambda(\pi)$ is equivalent to maximizing

$$D_{\text{out}} + (D_{\text{in}} - D_{\text{out}}) \sum_{m=1}^M w_m(\pi)^2 - D_{\text{in}} \sum_{y \in G} \pi(y)^2,$$

or, equivalently (since $\lambda > 0$ and constants do not affect argmax), to minimizing

$$F(\pi) := (D_{\text{out}} - D_{\text{in}}) \sum_{m=1}^M w_m(\pi)^2 + D_{\text{in}} \sum_{y \in G} \pi(y)^2, \tag{7}$$

where $D_{\text{out}} - D_{\text{in}} > 0$ and $D_{\text{in}} \geq 0$.

### D.4    STEP 3: COVERAGE STRUCTURE AND ENTROPY OF OPTIMAL POLICIES ON $G$

Having reduced both objectives to $\Delta(G)$ (Lemma 2 and Lemma 3), we now compare their optimal solutions. First, Lemma 2 immediately implies:

**Corollary 1** (Flat optimal set for correctness-only objective)**.** *Under $G \neq \emptyset$, any $\pi \in \Delta(G)$ satisfies $J_0(\pi) = 1$ and is a global maximizer of $J_0$. Hence $J_0$ is indifferent to how probability mass is distributed within $G$.*

In contrast, $J_\lambda$ has a nontrivial preference both at the mode level and within each mode, captured by the quadratic form $F(\pi)$ in equation 7.

We first show that, for any fixed mode-mass vector $w$, $J_\lambda$ is maximized by making the policy uniform within each mode.

**Lemma 5** (Within each mode, optimal policies are uniform)**.** *Fix $w \in \Delta_M$ and consider the set*

$$\mathcal{P}(w) := \big\{\pi \in \Delta(G) : w_m(\pi) = w_m \text{ for all } m\big\}.$$

*Under Assumptions 1 and 2, among all $\pi \in \mathcal{P}(w)$, $J_\lambda(\pi)$ is maximized (equivalently, $F(\pi)$ is minimized) by policies that are uniform within each mode:*

$$\pi(y) = \frac{w_m}{N_m} \quad \text{for all } y \in G_m.$$

*If $D_{\text{in}} > 0$, this choice is unique in $\mathcal{P}(w)$; if $D_{\text{in}} = 0$, $J_\lambda$ is independent of the within-mode distribution.*

*Proof.* For fixed $w$, the term $\sum_m w_m(\pi)^2$ in equation 7 is constant over $\mathcal{P}(w)$. Thus minimizing $F(\pi)$ over $\mathcal{P}(w)$ is equivalent to minimizing $\sum_{y \in G} \pi(y)^2$ over $\mathcal{P}(w)$.

Within each mode $G_m$, this is the classical problem of minimizing a sum of squares subject to a linear constraint:

$$\min\Big\{ \sum_{y \in G_m} \pi(y)^2 : \sum_{y \in G_m} \pi(y) = w_m, \ \pi(y) \geq 0 \Big\},$$

whose unique solution (when $w_m > 0$) is the uniform allocation $\pi(y) = w_m/N_m$ for all $y \in G_m$. If $D_{\text{in}} > 0$, this sum of squares enters $F(\pi)$ with a strictly positive coefficient, so any deviation from uniformity strictly increases $F(\pi)$. If $D_{\text{in}} = 0$, the intra-mode term vanishes from $F(\pi)$, and $F(\pi)$ (hence $J_\lambda$) depends only on $w$ and not on the within-mode distribution. $\qquad \square$

By Lemma 5, any global maximizer of $J_\lambda$ on $\Delta(G)$ must be uniform within each mode. We may therefore restrict attention to policies of the form

$$\pi(y) = \frac{w_m}{N_m} \quad \text{if } y \in G_m,$$

parameterized solely by $w \in \Delta_M$. Substituting this structure into equation 6 yields a purely mode-level objective.

Indeed, for such a $\pi$,

$$\sum_{y \in G} \pi(y)^2 = \sum_{m=1}^{M} \sum_{y \in G_m} \left( \tfrac{w_m}{N_m} \right)^2 = \sum_{m=1}^{M} \frac{w_m^2}{N_m}.$$

Plugging this into Lemma 4 gives

$$J_\lambda(\pi) = 1 + \lambda \Big[ D_{\text{out}} + (D_{\text{in}} - D_{\text{out}}) \sum_{m=1}^{M} w_m^2 - D_{\text{in}} \sum_{m=1}^{M} \frac{w_m^2}{N_m} \Big]$$

$$= 1 + \lambda \Big[ D_{\text{out}} - \sum_{m=1}^{M} a_m w_m^2 \Big], \tag{8}$$

where we define

$$a_m := (D_{\text{out}} - D_{\text{in}}) + \frac{D_{\text{in}}}{N_m} > 0.$$

Thus, among mode-wise uniform policies, maximizing $J_\lambda$ is equivalent to minimizing

$$\sum_{m=1}^{M} a_m w_m^2 \quad \text{subject to } w \in \Delta_M.$$

**Theorem 1** (Similarity reward selects high-coverage policies on $G$). *Suppose $G \neq \emptyset$, Assumptions 1 and 2 hold with $M \geq 2$, and $\lambda$ satisfies equation 5. Let $\pi^{(\lambda)}$ be any global maximizer of $J_\lambda$ over $\Delta(G)$, and let $w^{(\lambda)} := w(\pi^{(\lambda)})$ be its mode-mass vector. Then:*

*1. $\pi^{(\lambda)}$ is uniform within each mode:*

$$\pi^{(\lambda)}(y) = \frac{w_m^{(\lambda)}}{N_m} \quad \text{for all } y \in G_m, \ m = 1, \dots, M.$$

*2. $w^{(\lambda)}$ is the unique minimizer of $\sum_m a_m w_m^2$ over $\Delta_M$:*

$$w_m^{(\lambda)} = \frac{a_m^{-1}}{\sum_{j=1}^{M} a_j^{-1}}, \quad m = 1, \dots, M,$$

*where $a_m > 0$ is defined above. In particular, every mode receives strictly positive probability:*

$$w_m^{(\lambda)} > 0 \quad \text{for all } m = 1, \dots, M.$$

3. *If, in addition, $D_{\text{in}} > 0$ and all modes have equal size $N_m \equiv N/M$, then all $a_m$ coincide, $w^{(\lambda)}$ is uniform on modes, and $\pi^{(\lambda)}$ is the uniform distribution over all correct trajectories:*

$$\pi^{(\lambda)}(y) = \frac{1}{|G|} \quad \text{for all } y \in G.$$

*In this symmetric case, $\pi^{(\lambda)}$ maximizes the full policy entropy*

$$H(\pi) := -\sum_{y \in G} \pi(y) \log \pi(y),$$

*achieving $H(\pi^{(\lambda)}) = \log |G|$, the largest possible entropy on $\Delta(G)$.*

*In contrast, any $\pi^{(0)} \in \Delta(G)$ is optimal for $J_0$, including degenerate solutions that concentrate on a single mode (with mode-level entropy $0$ and very low trajectory-level entropy). Thus $J_\lambda$ provably selects high-coverage policies within $G$, using all modes and being uniform within each mode, and, under mild symmetry, the maximum-entropy fully uniform policy, while $J_0$ does not enforce any coverage.*

*Proof.* (i) By Lemma 3, any global maximizer $\pi^{(\lambda)}$ lies in $\Delta(G)$. Lemma 5 then implies that any maximizer must be uniform within each mode, so $\pi^{(\lambda)}$ has the stated form.

(ii) For such mode-wise uniform policies, equation 8 shows that maximizing $J_\lambda$ is equivalent to minimizing $\sum_m a_m w_m^2$ over $w \in \Delta_M$. The function $w \mapsto \sum_m a_m w_m^2$ is strictly convex on $\Delta_M$ because each $a_m > 0$, and the constraint set $\Delta_M$ is convex and compact. Using Lagrange multipliers for the equality constraint $\sum_m w_m = 1$, the unique minimizer satisfies

$$2a_m w_m^{(\lambda)} + \mu = 0 \quad \text{for all } m,$$

for some scalar $\mu$, which yields

$$w_m^{(\lambda)} = \frac{-\mu}{2a_m} = \frac{a_m^{-1}}{\sum_{j=1}^{M} a_j^{-1}}, \quad m = 1, \ldots, M.$$

All $a_m > 0$, so all $w_m^{(\lambda)} > 0$.

(iii) If $D_{\text{in}} > 0$ and $N_m \equiv N/M$ for all $m$, then

$$a_m = (D_{\text{out}} - D_{\text{in}}) + \frac{D_{\text{in}}}{N_m} = (D_{\text{out}} - D_{\text{in}}) + \frac{D_{\text{in}}}{N/M}$$

is the same for all $m$. Hence $w_m^{(\lambda)} = 1/M$ for all $m$, and

$$\pi^{(\lambda)}(y) = \frac{w_m^{(\lambda)}}{N_m} = \frac{1/M}{N/M} = \frac{1}{N} = \frac{1}{|G|}$$

for all $y \in G$, i.e., $\pi^{(\lambda)}$ is the uniform distribution over $G$.

Finally, the Shannon entropy $H(\pi)$ over $G$ is maximized on $\Delta(G)$ exactly by the uniform distribution, with value $\log |G|$. Thus $H(\pi^{(\lambda)}) = \log |G|$, the largest possible entropy on $\Delta(G)$. $\qquad\square$

**Summary.** Under the binary-reward setting and a piecewise-homogeneous similarity-gap structure on the correct set, we have shown that for any sufficiently small similarity weight $\lambda$:

(a) ( Lemma 2 and Lemma 3) Any global maximizer of $J_0$ or $J_\lambda$ places all its probability mass on the correct set $G$; i.e., both objectives "converge to the answer-correct set" in a static sense.

(b) (Lemma 1) The correctness-only objective $J_0$ is completely flat on $\Delta(G)$: every distribution over $G$ is globally optimal, including degenerate policies that collapse onto a single mode and assign zero probability to many correct trajectories.

    (c) (Theorem 1) In contrast, the similarity-augmented objective $J_\lambda$ has global maximizers that must put positive mass on *all* modes and are uniform within each mode, and in the symmetric case where all modes have equal size, it selects the uniform distribution over all correct trajectories, which maximizes the full policy entropy.

Consequently, if a training procedure converges to global maximizers of $J_0$ and $J_\lambda$, then in the binary setting with similarity gaps: both trainings converge to fully correct policies, but the similarity-augmented training *provably selects* strictly more dispersed solutions on the correct set $G$, using all modes and, under mild symmetry, the maximum-entropy policy—while correctness-only training does not favor coverage and may converge to highly collapsed solutions.

## E  USE OF LARGE LANGUAGE MODELS IN PREPARATION

We acknowledge the use of Large Language Models (LLMs) as assistants in the preparation of this manuscript. Their role included refining phrasing and improving the clarity of the text, as well as assisting with programming tasks such as code generation and debugging for our experiments. The authors critically reviewed, edited, and verified all LLM-generated content for accuracy and appropriateness, and take full responsibility for the final content of this paper.

