# OpenReview forum: "On the Evolution of Language Models without Labels: Majority Drives Selection, Novelty Promotes Variation"
_ICLR.cc/2026/Conference — Submitted to ICLR 2026_

### Official Review · Reviewer_5QAG · 2025-11-01

**Soundness:** 3
**Presentation:** 3
**Contribution:** 3
**Rating:** 6
**Confidence:** 3

**Summary:**

This paper proposes EVOL-RL, a label-free reinforcement learning framework that helps language models improve themselves without external rewards or ground-truth labels. The authors show that existing self-training methods relying on confidence or majority voting often cause entropy collapse—models become over-confident, lose diversity, and generalize poorly. EVOL-RL addresses this by combining a majority-based stability signal with a novelty-aware reward that values reasoning diversity, mirroring the evolutionary balance between selection and variation. On GRPO, the method consistently boosts pass@1 and pass@16 accuracy across math and reasoning benchmarks and improves out-of-domain generalization.

**Strengths:**

The paper has several clear strengths. It’s very well-motivated — the authors clearly identify why current label-free learning approaches tend to collapse in diversity, and the diagnosis of “entropy collapse” feels both intuitive and empirically grounded. The proposed solution, combining majority-based stability with a novelty-driven exploration term, is conceptually straightfoward and surprisingly simple to implement. The experiments are strong and consistent across datasets and model sizes, showing meaningful gains in both pass@1 and pass@16 accuracy, which indicates that the method improves reasoning reliability without sacrificing diversity. I also appreciated the inclusion of detailed training-dynamics plots and ablations, which make it much easier to understand why the approach works rather than just showing numerical improvements.

**Weaknesses:**

Although the paper is strong overall, several aspects could be clearer or more complete.  The evaluation is solid but somewhat narrow — almost all experiments are done with the Qwen3 family, so it’s not clear whether the proposed reward structure would transfer cleanly to other models of different base model reasoning capabilities like Llama.

It would also help to include additional baselines beyond TTRL, such as entropy-regularized self-training, to better situate the improvement.

Finally, while the results and visualizations are convincing, the paper does not quantify the compute or runtime cost of computing similarities for each prompt, which would matter for scaling up.

**Questions:**

See weakness.

---

> ### Author Response · Authors · 2025-11-19
> **Response to Reviewer 5QAG**
>
> We sincerely thank the reviewer for the strong endorsement of our paper and the constructive suggestions regarding model generality, baselines, and compute costs. We have addressed all these points with new experiments and detailed analyses in the revision.
>
> ## 1. Concern regarding transferability to other model families (e.g., Llama).
> To address the transferability concern, we added new experiments in Appendix B.1 using **OctoThinker-8B-Hybrid-Base (a Llama-3.1-8B derivative optimized for reasoning)**. We selected this model because **standard Llama-3 base models often struggle to acquire reasoning directly via RL [1]**.
>
> The results (see table below) strongly confirm that EVOL-RL generalizes effectively. While the TTRL baseline suffers from entropy collapse—improving pass@1 but degrading pass@16 on almost all other tasks—EVOL-RL consistently prevents this collapse . Crucially, EVOL-RL delivers massive improvements on out-of-domain benchmarks where TTRL fails, such as **+19.5%** pass@16 on AIME24 and **+11.4%** pass@16 on AIME25.
>
> | Model | MATH | AIME24 | AIME25 | AMC | GPQA |
> |-------|------|-----|-----|-----|------|
> | Base | 33.8/79.8 | 1.5/13.5 | 1.3/13.9 | 16.2/56.9 | 26.3/85.7 |
> | TTRL | 63.8/76.4 | 2.8/10.8 | 2.1/11.0 | 27.9/54.7 | 31.9/71.5 |
> | **EVOL-RL** | **63.2/86.3** | **9.0/30.3** | **7.2/22.4** | **34.1/65.6** | **33.2/85.7** |
> | Δ | -0.6/+9.9 | +6.2/+19.5 | +5.1/+11.4 | +6.2/+10.9 | +1.3/+14.2 |
>
> Note: Cell values denote pass@1 / pass@16.
>
>
> ## 2. Suggestion to include additional baselines (e.g., entropy-regularized self-training).
> We agree that comparing against entropy-regularized approaches is crucial. We expanded our evaluation in Appendix B.2 to include three additional baselines trained on MATH-500: EM-RL-Token, EM-RL-Sequence (Entropy Minimization) [2], and Self-Consistency [3,4].
>
> The results (summarized below for Qwen3-4B-Base) reveal that simple entropy regularization is insufficient. On AIME24, while EM-RL performs slightly better than TTRL, it still fails to improve pass@16 over the Base Model (31.3% vs 32.4%). This indicates that non-directional entropy regularization adds noise but cannot drive the discovery of new solution paths. In contrast, EVOL-RL provides the necessary directional guidance, achieving a **+15.2%** gain over the base model in pass@16 on AIME24 and **+12.0%** gain over the base model in pass@1 on AIME25.
>
>
> | Model | MATH | AIME24 | AIME25 | AMC | GPQA |
> |-------|------|--------|--------|-----|------|
> | Base | 67.4/89.6 | 10.0/32.4 | 5.5/30.0 | 39.3/75.2 | 34.4/85.6 |
> | TTRL | 75.4/86.9 | 12.1/23.2 | 6.8/28.6 | 42.5/75.2 | 36.5/81.4 |
> | EM-RL-Token | 76.0/90.6 | 12.5/31.3 | 10.5/30.8 | 46.6/77.7 | 36.8/82.6 |
> | EM-RL-Seq | 67.4/89.9 | 10.6/31.4 | 7.1/28.8 | 39.6/73.7 | 34.5/86.2 |
> | Self-Consistency | 76.0/89.7 | 12.5/30.4 | 10.4/33.6 | 48.1/78.1 | 35.9/81.2 |
> | **EVOL-RL** | **80.0/93.3** | **20.7/47.6** | **17.5/39.9** | **51.4/80.3** | **37.2/88.7** |
>
>
> Note: Cell values denote pass@1 / pass@16.
>
> ## 3. Request to quantify compute and runtime costs.
> To address scalability concerns, we added a detailed runtime analysis in Appendix B.5 (Figure 7), tracking the wall-clock time for Novelty Reward Calculation vs. the standard Training Process.
> * **Absolute Cost**: 	As reasoning chains grow longer during training, the time required to encode them with the embedding model increases, stabilizing at ~100 seconds per step.
> * **Relative Cost**: Crucially, the training process is dominated by the generation of long CoT sequences and model updates. The novelty reward calculation **remains a minor overhead relative to the total computational load**. We conclude that the cost scales predictably and is a small fraction of the training budget, making it a viable solution for scaling up.
>
> **We once again thank the reviewer for these insightful suggestions, which have significantly improved the completeness of our paper. If the clarifications provided have resolved your concerns, we would appreciate it if you could consider a higher evaluation.**
>
> [1] Gandhi et al. "Cognitive behaviors that enable self-improving reasoners, or, four habits of highly effective stars"
>
> [2] Agarwal, Shivam, et al. "The unreasonable effectiveness of entropy minimization in llm reasoning.”
>
> [3] Huang, Jiaxin, et al. "Large language models can self-improve."
>
> [4] Wang, Yizhong, et al. "Self-instruct: Aligning language models with self-generated instructions."

---

> ### Author Response · Authors · 2025-11-24
>
> Dear Reviewer 5QAG,
>
> Thank you again for your thoughtful feedback. We have carefully addressed your concerns and added substantial new experiments to strengthen our submission. If convenient, we would greatly appreciate a brief follow-up—whether the responses fully resolve your concerns or if there are still remaining points to clarify. We are very happy to engage further in discussion.
>
> Thank you for your time and consideration.

---

### Official Review · Reviewer_UJzh · 2025-11-03

**Soundness:** 3
**Presentation:** 3
**Contribution:** 2
**Rating:** 4
**Confidence:** 4

**Summary:**

This paper introduces EVOL-RL, a RL framework that integrates majority voting and semantic novelty into the reward function. The core idea is to encourage the model to produce semantically novel yet self-consistent responses, extending majority voting to mitigate diversity collapse and mode collapse. Experimental results on Qwen3-4B-Base and Qwen3-8B-Base models demonstrate the effectiveness and robustness of the proposed method.

**Strengths:**

- The proposed method is clearly present and easy to follow.
- The experimental results show that EVOL-RL actually achieves notably improvement in pass@16 accuracy, which is a big plus.

**Weaknesses:**

- EVOL-RL primarily builds upon self-consistency-based reward, adding an auxiliary objective to promote response diversity. However, the method remains fundamentally dominated by self-consistency. In my view, this contributes little to improving the model's reasoning capability. For example, when tackling high-difficulty problems, most LLM responses are incorrect, meaning the reward signal derived from majority voting is unreliable. Consequently, the method cannot effectively guide the model toward the correct reasoning trajectory.
- Using cosine similarity in the embedding space to measure semantic novelty is a rather coarse approach. For high-dimensional embeddings, cosine similarity may not provide enough discrimination. In addition, it only reflects novelty at the semantic level, not the logical novelty of the reasoning process, and the latter is what truly matters for reasoning exploration.
- Validation on small-scale models introduces considerable uncertainty. I suggest that the authors include experiments on larger models (e.g., 32B parameters), which would strengthen the reliability of the results and better demonstrate the method’s effectiveness.
- The proposed Novelty Reward could likely be integrated into the RLVR framework (using GT reward instead of major voting reward) as well. I recommend adding an additional experiment combining RLVR + Novelty Reward. If this setup also yields a clear improvement in pass@16, it would further support the generality and effectiveness of the proposed novelty-based reward design.

Overall, I take a cautious approach and give a borderline reject recommendation. I look forward to the authors' response.

**Questions:**

See above

---

> ### Author Response · Authors · 2025-11-19
> **Response to Reviewer UJzh**
>
> We sincerely thank the reviewer for the constructive and critical feedback. We have carefully addressed your concerns with new experiments and detailed analyses, which we believe significantly strengthen the paper.
>
> ## 1. Concern regarding the validity of the self-consistency paradigm and effectiveness on hard problems.
> We thank the reviewer for this critical question regarding the reliability of majority voting on difficult tasks.
> * **Validity of the Label-Free Paradigm**: The label-free setting is crucial for scaling LLMs where ground truth is missing. As analyzed theoretically in **Section 4.2 of the highly-cited TTRL paper (NeurIPS 2025) [1]**, majority-based RL can improve models even when the voting signal is noisy or imperfect. The wide recognition of the TTRL confirms that this paradigm is mathematically sound and empirically viable.
> * **EVOL-RL Effectively Solves Hard Problems**: While TTRL establishes the paradigm, our work addresses its critical failure mode: **Entropy Collapse**. Our results empirically refute the concern that the method cannot guide the model on hard tasks due to unreliable signals.
>     * **Performance on Hard Tasks**: On the challenging **AIME24** benchmark (where the base model has only **10.0%** accuracy and the majority signal is highly noisy), EVOL-RL successfully doubled the performance to **20.6%**, whereas the baseline collapsed.
>     * **Closing the Gap with Supervised RL**: Notably, as shown in our experiments, the performance gap between our **label-free** EVOL-RL (Table 1) and the **RLVR (with ground truth)** (Figure 4) is surprisingly small. This demonstrates that our novelty-guided exploration effectively compensates for the lack of ground truth, enabling robust learning even in high-difficulty, label-free scenarios.
>
> ## 2. Concern that cosine similarity is too coarse to capture logical novelty.
> We agree that distinguishing logical novelty from semantic variation is a central challenge, but there are no direct ways to measure the logical similarity. To verify if our embedding-based metric captures logical differences, we added a detailed case study in **Appendix B.6 (Figure 8)**.
>
>
> * Case Study Results: We generated 8 reasoning paths for a single AIME25 problem. The resulting similarity matrix showed distinct, well-separated clusters that aligned perfectly with the logical outcomes.
> * High Discrimination: The 6 rollouts that used the same logic to reach the answer "19" formed a tight cluster, which was clearly distinguishable from paths reaching "31" or "7221". This empirically confirms that in the context of CoT, the embedding space effectively clusters paths by underlying logic, making cosine similarity a valid proxy for rewarding genuine exploration.
> ## 3. Suggestion to include experiments on larger models.
> We thank the reviewer for this suggestion. We sincerely apologize that running experiments on 32B-parameter models is not feasible given our current computational resources and the limited rebuttal period. We hope the reviewer can understand this constraint. However, to provide the strongest possible evidence for our method's reliability and effectiveness, we have done the following:
>
> **New Model Architecture**: In addition to the Qwen3-4B-Base and 8B-Base results in the main paper, we have added experiments on **OctoThinker-8B-Hybrid-Base** in the revision (**Appendix B.1**). The results on OctoThinker confirm significant improvements over the TTRL baseline, consistent with our other findings.
>
> We believe the consistent positive results across three distinct models provide strong evidence for the validity of our approach.
>
> To ensure our work is fully transparent and reproducible, we have provided an anonymous link to our codebase and welcome the community to validate our findings on larger-scale models.
>
> **Anonymous Code**: https://anonymous.4open.science/r/submission8707
>
> ## 4. Recommendation to integrate Novelty Reward with RLVR (Ground Truth).
>
> * **Experiment Confirmation**: We are pleased to report that this experiment was included in our initial submission. As presented in Section 4.4 and Figure 4, we combined our novelty reward with the standard GRPO with ground truth (RLVR).
> * **Results**: This combination yielded clear improvements in both pass@1 and pass@16 performance. This confirms that our novelty mechanism is a general-purpose method for boosting reasoning diversity and performance, effective even when ground truth signals are available.
>
> **We once again thank the reviewer for these insightful comments, which have helped us improve the robustness of our evaluation. We are happy to answer any further questions. If our response has resolved your concerns, we kindly ask that you consider raising your score.**
>
> [1] Zuo et al. "TTRL: Test-time reinforcement learning." NeurIPS 2025.

---

> ### Author Response · Authors · 2025-11-24
>
> Dear Reviewer UJzh,
>
> Thank you again for your thoughtful feedback. We have carefully addressed your concerns and added substantial new experiments to strengthen our submission. If convenient, we would greatly appreciate a brief follow-up—whether the responses fully resolve your concerns or if there are still remaining points to clarify. We are very happy to engage further in discussion.
>
> Thank you for your time and consideration.

---

### Official Review · Reviewer_zE94 · 2025-11-09

**Soundness:** 3
**Presentation:** 3
**Contribution:** 3
**Rating:** 4
**Confidence:** 5

**Summary:**

This paper introduces a straightforward idea regarding label free RL training. It solves the entropy collapse issue from previous label free training approaches.

Inspired by biological evolution, they propose a novel label free RL training design which balances two aspects of the RL training: selection (picking good answers) and variation (trying new things).

Based on this intuition, they designed EVOL-RL, a practical system that complements the majority-vote reward with a "novelty-aware reward" for new reasoning paths. This approach proved highly effective, avoiding performance decline and enabling models to maintain longer, more informative chains of thought. Empirically, this design brings state-of-the-art results, and can improve generalization to unseen problems where prior methods fail.

**Strengths:**

- This paper exposes and directly fixes "entropy collapse," a critical limitation of previous label-free RL methods that causes models to converge on a single, repetitive solution and lose reasoning diversity.

- The EVOL-RL method can improve non-trivially over previous TTRL by rewarding additional "novelty," which encourages to maintain longer, more informative, and varied chains of thought instead of just finding the shortest, most common answer.

- It also shows strong generalization, improving on unseen tasks it wasn't trained on.

**Weaknesses:**

- Will this design work on other base series? There are concerns that the Qwen base model has trained on massive reasoning data during its annealing and SFT stage which could potentially have "evaluation leakage". It makes it hard to judge if the current Evol-RL design will only work on a Qwen kind of base model or it can have broader generalization implications for other base models.

- The "novelty-aware reward" isn't free. It requires a separate process to calculate the semantic difference between solutions, adding higher computational cost and additional engineering complexity to the training setup.

- There are other ways to improve a model's entropy, for example changing the temperature to encourage more diverse and stochastic output and etc. The authors didn't address other existing solutions to address the entropy collapse issue [1] - [6].

- The system measures novelty using semantic similarity in an embedding space. This is a proxy for real, human-like creativity and may not always capture what makes a reasoning path genuinely inventive.


[1] Dai et al. CDE: Curiosity-Driven Exploration for Efficient Reinforcement Learning in Large Language Models. 2025

[2] Li et al. Jointly Reinforcing Diversity and Quality in Language Model Generations. 2025

[3] Cui et al. The Entropy Mechanism of Reinforcement Learning for Reasoning Language Models. 2025

[4] Park et al. Clip-Low Increases Entropy and Clip-High Decreases Entropy in Reinforcement Learning of Large Language Models. 2025

[5] Shen, H. On Entropy Control in LLM-RL Algorithms. 2025

[6] Zhao et al. The Majority is not always right: RL training for solution aggregation. 2025

**Questions:**

See the above Weaknesses. Mainly including:

Validating the design on other base models.

More detailed computation cost analysis.

Better justifying the position of this work compared to other existing work that attempts to resolve entropy collapse and novelty encouragement.

---

> ### Author Response · Authors · 2025-11-19
> **Response to Reviewer zE94**
>
> We sincerely thank the reviewer for the constructive and critical feedback. We have addressed these concerns with new experiments and detailed analyses.
>
>
> ## 1. Generalization to Other Base Models
>
> To explicitly test transferability of EVOL-RL, we have added a new experiment in Appendix B.1 using **OctoThinker-8B-Hybrid-Base trained on MATH-500**.
>
> * Model Choice: We selected OctoThinker (mid-trained on Llama-3.1-8B Base model) because, as noted in [7], standard Llama-3 base models struggle to acquire reasoning capabilities directly from RL. OctoThinker's mid-training stage makes it amenable to RL, making it the ideal testbed.
>
> * Results: The results (see Table below) directly refute the concern. On this Llama-based model, the TTRL baseline suffers severe entropy collapse, actually degrading pass@16 performance on AIME24, AIME25, and GPQA. In contrast, **EVOL-RL robustly prevents this collapse**, achieving massive gains in pass@16 (e.g., **+19.5%** on AIME24 and **+11.4%** on AIME25 over TTRL) and significant pass@1 improvements. This confirms that our design principles are model-agnostic.
> | Model | MATH | AIME24 | AIME25 | AMC | GPQA |
> |-------|------|-----|-----|-----|------|
> | Base | 33.8/79.8 | 1.5/13.5 | 1.3/13.9 | 16.2/56.9 | 26.3/85.7 |
> | TTRL | 63.8/76.4 | 2.8/10.8 | 2.1/11.0 | 27.9/54.7 | 31.9/71.5 |
> | **EVOL-RL** | **63.2/86.3** | **9.0/30.3** | **7.2/22.4** | **34.1/65.6** | **33.2/85.7** |
> | Δ | -0.6/+9.9 | +6.2/+19.5 | +5.1/+11.4 | +6.2/+10.9 | +1.3/+14.2 |
>
> Note: Cell values denote pass@1 / pass@16.
>
>
> ##  2. Cost and Complexity of Novelty-Aware Reward
>
> We thank the reviewer for this practical point. We have analyzed both cost and complexity:
>
> * **Computational Cost (Relative Overhead):** To rigorously quantify this, we added a runtime analysis in Appendix B.5 (Figure 7), plotting the wall-clock time per step for the Novelty Reward Calculation versus the Total Training Process. While the reward cost scales with response length (requiring embedding of longer sequences), the Novelty Reward Calculation (red curve) remains a small portion of the overall runtime, stabilizing around 100 seconds.
>
> * **Engineering Complexity:** The process is streamlined, requiring only: (1) inference on an off-the-shelf embedding model (no training), and (2) standard vectorized cosine similarity ($O(B \cdot N^2)$). This is a high-return, low-complexity solution.
>
> ## 3. Comparison with Entropy-Regularized Baselines
>
> We thank the reviewer for these highly relevant citations, which help us situate our contribution.
>
> First, we must clarify that with the exception of [3], all other cited papers [1, 2, 4, 5, 6] are concurrent work (arXiv Sept. 2025). We added a discussion to Related Work to clarify distinctions:
> * **Why a simple Entropy Bonus [3-5] is Insufficient:** Our ablation (Sec 4.5) shows that adding generic entropy loss (TTRL+Entropy) raises policy entropy but fails to solve performance degradation. An entropy bonus is non-directional; it encourages randomness but lacks guidance to escape the majority trap.
> * **Our Solution is a Directional Reward:** EVOL-RL's core contribution is not just adding exploration, but adding a directional novelty reward that provides a targeted selection pressure to escape this trap. The analysis of the effectiveness of this reward design is provided in Sec. 3.3.
> * **Others:** CDE [1] operates in a supervised RLVR setting (we are label-free), and Darling [2] requires training a separate evaluator (we only used a pre-trained embedding model). We have cited [6] in another relevant section, as it does not focus on exploration.
>
> ## 4. Validity of Semantic Similarity as a Proxy for Novelty
>
> We thank the reviewer for this insightful comment regarding the proxy's validity. To address this, we added a case study in Appendix B.6 (Figure 8), where we computed the reasoning path similarity matrix for 8 rollouts on a single AIME25 problem.
>
> * **The Proxy is Valid:** The results provide strong empirical evidence. The matrix shows distinct, high-similarity clusters that align perfectly with the reasoning logic. Specifically, the 6 rollouts that produced the same final answer ("19") have an extremely high intra-cluster similarity. Crucially, these paths are **semantically distinct** (showing clear separation in the matrix) from the paths that reached different answers ("31" and "7221").
> * This demonstrates that the embedding space **does effectively capture the semantic essence of the reasoning path**, successfully clustering different logics and distinguishing between different reasoning trajectories.
>
> **We thank the reviewer again for these excellent suggestions. We hope these responses fully address your concerns. We’re here to help with any other questions you may have. If you’re satisfied with the support provided, we would be grateful if you could rate us positively.**
>
> [7] Gandhi et al. "Cognitive behaviors that enable self-improving reasoners, or, four habits of highly effective stars"

---

> ### Author Response · Authors · 2025-11-24
>
> Dear Reviewer zE94,
>
> Thank you again for your thoughtful feedback. We have carefully addressed your concerns and added substantial new experiments to strengthen our submission. If convenient, we would greatly appreciate a brief follow-up—whether the responses fully resolve your concerns or if there are still remaining points to clarify. We are very happy to engage further in discussion.
>
> Thank you for your time and consideration.

---

### Official Review · Reviewer_zAS9 · 2025-11-10

**Soundness:** 2
**Presentation:** 3
**Contribution:** 2
**Rating:** 4
**Confidence:** 3

**Summary:**

This paper studies label‑free self‑improvement of large language models (LLMs). The authors argue that naïve self-training tends to collapse entropy (shorter, less diverse outputs). They propose EVOL-RL, combining (1) majority-voted selection for stability and (2) novelty-aware rewards to encourage variation. The method uses a semantic-space novelty reward with an entropy regularizer and asymmetric clipping in a generalized RL framework. Experiments on reasoning tasks (AIME24→AIME25) show substantial gains in Pass@1 and Pass@k over majority-only baselines.

**Strengths:**

* Combining majority selection and novelty reward is intuitive, easy to implement, and empirically effective.
* Significant empirical improvements on challenging reasoning tasks, with out-of-domain transfer tested.
* Clear presentation and reproducible implementation

**Weaknesses:**

* Majority selection + novelty/diversity reward is conceptually straightforward and has analogues in prior work (self-consistency, diversity-encouraging RL). The paper should clarify distinctions and contributions relative to prior diversity-aware self-training approaches. [1][2]
* Embedding-based novelty reward may require more computation load. Long-term behavior over many self-improvement iterations is unclear; does novelty continue to help or collapse eventually?
* The current reward, while balancing selection and variation, may overly penalize semantically similar reasoning paths that produce an incorrect final answer. For example, a reasoning trajectory almost identical to a majority-correct solution but failing on a minor step would receive a low reward. This strict focus on final correctness could discourage exploration of "almost correct" reasoning strategies, limiting the model’s ability to refine nuanced problem-solving behaviors.
* The paper appears to use test-set examples during self-training (e.g., AIME24→AIME25 transfer).

[1] Pang J C, Wang P, Li K, et al. Language model self-improvement by reinforcement learning contemplation[J]. arXiv preprint arXiv:2305.14483, 2023.
[2] Shuvendu Roy, Hossein Hajimirsadeghi, ta al. You Need Reasoning to Learn Reasoning: The Limitations of Label-Free RL in Weak Base Models[J]. arXiv preprint arXiv:2511.04902, 2025.

**Questions:**

Please refer to the weakness.

---

> ### Author Response · Authors · 2025-11-19
> **Response to Reviewer zAS9**
>
> We sincerely thank the reviewer for the constructive and insightful feedback. We have carefully addressed all concerns. We hope these discussions resolve your concerns.
>
> ## 1. Distinction from Prior Diversity-Aware Approaches [1][2]
>
> We have expanded Section 2 to explicitly contrast our approach with the cited works. We would like to note that [2] is concurrent work appearing after our submission.
>
> Specifically, we now contrast (1) SIRLC’s single-sample self-evaluation rewards with our group-level majority-plus-novelty objective, and (2) CuMa’s curriculum-based stabilization techniques with our evolutionary formulation that directly resolves the majority trap via semantic-novelty rewards. These additions clarify our unique contribution relative to prior diversity-aware or label-free self-training approaches.
>
> ## 2. Computational Load and Long-Term Stability
>
> * **On Computational Load**: To rigorously quantify this, we have added a runtime analysis in Appendix B.5 (Figure 7), plotting the wall-clock time per step for the Novelty Reward Calculation (Red curve) versus the standard Training Process (Blue curve).
>     * Scaling: 	The reward calculation cost is not constant; it increases as the model learns to generate longer reasoning chains (requiring embedding of longer sequences).
>     * Low Relative Overhead: However, as shown in the figure, the generation + updates process is computationally expensive and dominates the runtime. In comparison, the novelty reward calculation remains a **relatively small overhead**. We conclude that this modest relative cost is a favorable trade-off for the >3x performance gains.
>
> * **On Long-Term Stability**: This is addressed in Section 4.4 (lines 416-426). To make it clearer, we updated Figure 3 with **a longer 300-step training run on MATH-500**. The results on MATH-500 and AIME24 show that after the "evolving point" ($\sim$ step 150), all key metrics—entropy, response length, and accuracy—stabilize and fluctuate within a small range rather than rising indefinitely. This empirically confirms EVOL-RL finds a healthy, high-performing equilibrium.
>
> ## 3. Reward Penalty on "Almost Correct" Reasoning
>
> The reviewer correctly points out that a strict focus on correctness might discourage "almost correct" strategies. However, our reward mechanism handles this nuance via the **intra-group mean similarity ($\bar{s}_i$) term** in our formula: $u_i = 1 - (\alpha \bar{s}_i + (1-\alpha) m_i)$. It uses group-level similarity to protect rare exploratory errors while penalizing common systematic failures. We discuss two cases for the "almost correct" failure.
> * **Case A: An Exploration (Rare Failure).** If the "almost correct" error is a rare, occasional mistake  (i.e., a valid attempt at exploration), it will be semantically dissimilar to other failure modes in the minority group. This results in a LOW $\bar{s}_i$. In our formula, a low $\bar{s}_i$ counteracts the high global similarity ($m_i$), resulting in a higher novelty score. This mitigates the penalty, protecting valid exploration. Most “almost correct” failures fall into this category, unless the model consistently makes the same mistake.
> * **Case B: The Systematic Trap (Common Failure).** If the error represents a high-frequency failure mode (e.g., a common misconception), it will be similar to many other minority solutions. This results in a HIGH $\bar{s}_i$, leading to a low novelty score and a heavy penalty.
>  We have added this analysis to Appendix C.1.
>
> ## 4. Training on Test Sets (Data Leakage Concern)
>
> The reviewer's observation is correct: we use test sets as a source of unlabeled training data. We emphasize two key points:
> * **Strictly Label-Free: No ground-truth labels or answers were ever used.** The only supervisory signal is the model's own self-generated "majority vote," preventing any label leakage.
> * **Following the Established TTRL Paradigm**: This setup is chosen to directly compare against the **Test-Time Reinforcement Learning (TTRL) paradigm [3]**. This paradigm (accepted at NeurIPS 2025) simulates deployment to a new, unknown domain where the model must self-adapt using only new, unlabeled data. Adhering to this setup is crucial for fair comparison. Finally, to prove our method is not limited to this paradigm, we also show in Table 1 that EVOL-RL performs equally well on the large-scale, traditional MATH-TRAIN set.
>
> **We thank the reviewer again for these excellent suggestions, which have significantly strengthened our paper's positioning and analysis. If you’ve got more questions, we’re always here to help. And if everything’s sorted out, maybe you could consider giving us a higher rating?**
>
> [1] Pang J C, Wang P, Li K, et al. Language model self-improvement by reinforcement learning contemplation。
>
> [2] Shuvendu Roy, Hossein Hajimirsadeghi, ta al. You Need Reasoning to Learn Reasoning: The Limitations of Label-Free RL in Weak Base Models.
>
> [3] Zuo et al. "TTRL: Test-time reinforcement learning." NeurIPS 2025.

---

> ### Author Response · Authors · 2025-11-24
>
> Dear Reviewer zAS9,
>
> Thank you again for your thoughtful feedback. We have carefully addressed your concerns and added substantial new experiments to strengthen our submission. If convenient, we would greatly appreciate a brief follow-up—whether the responses fully resolve your concerns or if there are still remaining points to clarify. We are very happy to engage further in discussion.
>
> Thank you for your time and consideration.

---

### Author Response · Authors · 2025-11-19
**Summary of Revision Changes**

To address the reviewers' feedback and improve the paper’s clarity and robustness, we made the following updates:

1. **Extended Training Dynamics (Figure 3 & Figure 5):**

   * We extended the training duration on **MATH-500** to ensure stability.
   * The results show that EVOL-RL remains stable for up to 300 steps, while the baseline models degrade or plateau.

2. **Expanded Related Work (Section 2):**

   * We added a more detailed comparison with concurrent entropy-minimization and label-free methods.
   * This clarifies how our directional novelty reward differs from non-directional exploration or generic entropy bonuses.

3. **Theoretical Analysis of Reward Formulation (Appendix C.1):**

   * We added an analysis explaining how the reward handles **“almost correct”** solutions.
   * The update clarifies how the method protects rare exploratory mistakes while penalizing common failure modes.

4. **Generalization to Different Architectures (Appendix B.1):**

   * We introduced experiments on **OctoThinker-8B-Hybrid-Base** to demonstrate transferability beyond the Qwen series.
   * EVOL-RL achieves a **+19.5%** pass@16 improvement on AIME24 over TTRL, confirming robust generalization.

5. **Comprehensive Baseline Comparisons (Appendix B.2):**

   * We added comparisons against **Self-Consistency**, **EM-RL-Token**, and **EM-RL-Sequence**.
   * Results show that simple entropy regularization fails to improve pass@16 on challenging benchmarks like AIME24.

6. **Runtime and Computational Cost Analysis (Appendix B.5):**

   * We added wall-clock analysis (Figure 7) comparing Novelty Reward Calculation to the training process.
   * The overhead of reward computation is minor relative to generating long reasoning chains.

7. **Validation of Semantic Novelty Metric (Appendix B.6):**

   * We included a case study visualizing the cosine similarity matrix of reasoning paths (Figure 8).
   * This demonstrates that the embedding-based metric effectively distinguishes different logical paths and serves as a valid proxy for reasoning novelty.

---

### Author Response · Authors · 2025-11-29
**Summary of Rebuttal and Revisions for Paper**

Dear Area Chair,

We recognize the immense workload ACs face during this period, so we have prepared this concise summary of our rebuttal and revisions to facilitate your decision-making process.

**Overall Assessment**: We are encouraged that the reviewers found our paper "strong overall" (Reviewer 5QAG), with "convincing results and visualizations" (Reviewer 5QAG), and acknowledged that we address a critical problem in label-free RL. Crucially, no factual errors or fundamental flaws in our methodology were identified. The concerns raised were primarily requests for broader verification (more models/baselines) and clarification (cost analysis, prior work distinctions), all of which we have fully addressed in the revision.

## **Summary of Our Rebuttal & New Evidence:**

To comprehensively address the reviewers' constructive suggestions, we have significantly expanded our evaluation and theoretical grounding:

### **1. Validating Generalization on Llama Architecture (Addressing Reviewers zE94, UJzh, 5QAG)**

**Our rebuttal**: We added experiments using OctoThinker-8B-Hybrid-Base (based on Llama-3.1).

**Result**: EVOL-RL successfully prevents entropy collapse on this architecture, achieving massive gains over the TTRL baseline (e.g., +19.5% pass@16 on AIME24), proving our method is model-agnostic.

### **2. Clarifying Computational Cost (Addressing Reviewers zAS9, zE94, 5QAG)**

**Our rebuttal**: We added a detailed runtime analysis (Figure 7 in Appendix).

**Result**: We demonstrate that while the reward calculation scales with response length, the total runtime is dominated by the standard generation/training process. The novelty reward remains a minor relative overhead, a highly favorable trade-off for the >3x performance gains.

### **3. Comparing with More Baselines Besides TTRL (Addressing Reviewers 5QAG)**

**Our rebuttal**: We added baselines for Entropy Maximization (EM-RL) and Self-Consistency.

**Result**: These methods fail to prevent diversity collapse on hard tasks (e.g., AIME24), confirming that our directional novelty reward is essential and superior to simple entropy regularization.

### **4. Validating the "Embedding-based Semantic Proxy" (Addressing Reviewers zE94, UJzh)**

**Our rebuttal**: We added a case study (Figure 8 in Appendix) analyzing the cosine similarity of the embedding of the rollouts from the same question.

**Result**: The results show that distinct embedding-based high-similarity clusters align well with the underlying logic and final answers, validating embedding space as an effective proxy for reasoning diversity.

### **5. Clarifying Paradigm & Prior Work (Addressing Reviewers zAS9, zE94, UJzh)**

**Our rebuttal**: We clarified our adherence to the established TTRL paradigm (NeurIPS 2025) to ensure fair comparison. We also expanded the Related Work section to explicitly distinguish our approach from concurrent works like SIRLC, CDE and CuMa.

**Note on Concurrent Work**: Importantly, one of the two papers mentioned by Reviewer zAS9 and **five of the six** papers mentioned by Reviewer zE94 appeared on arXiv after September 2025, classifying them as concurrent work relative to our submission. Nevertheless, we have included detailed discussions to clearly delineate our distinct contributions against these latest developments.

### **6. Added Theoretical Analysis (New Contribution)**

**To further elevate the rigor of our work, we added a formal theoretical analysis in Appendix D (Theorem 1).**

**Result**: We formally prove that with a correctness-only objective (like TTRL), the optimal policy can collapse onto a single response. In contrast, we show that under mild assumptions on similarity structure, our similarity-based novelty reward guarantees that the optimal policy must spread probability mass across multiple correct responses. This provides a theoretical foundation for why EVOL-RL stabilizes policy entropy and yields diverse solutions.

**We believe the manuscript is now substantially stronger, backed by both extensive empirical evidence and new theoretical proofs. We remain available for any further questions and hope this summary assists in your final assessment.**

Sincerely,

The Authors of Submission 8707

---

### Author Response · Authors · 2025-12-03
**Summary of Contributions and Experiments for Submission 8707**

**Dear Area Chair,**

To assist in the evaluation process, we have prepared a structured summary of our paper, **"ON THE EVOLUTION OF LANGUAGE MODELS WITHOUT LABELS,"** highlighting how **EVOL-RL** addresses the critical issue of entropy collapse in label-free learning. This summary integrates key findings from both the main text and the extensive supplementary experiments in the Appendix.

### **1. Core Problem & Solution**
* **The Problem (The Majority Trap):** We identify that existing label-free methods (e.g., TTRL) relying solely on **Majority Voting** induce **"Entropy Collapse."** While Pass@1 may marginally improve, the policy entropy drops to near-zero, response lengths shorten, and Pass@N (diversity) degrades significantly.
* **The Solution (EVOL-RL):** Inspired by evolutionary principles, we propose a dual-signal reward framework:
    * **Selection:** Uses majority voting to anchor correctness.
    * **Variation:** Introduces a **Novelty Reward** (based on reasoning trace semantic similarity) to incentivize diverse reasoning paths and penalize repetitive modes.

### **2. Systematic Experimental Evaluation**
Our experiments are comprehensive, covering multiple scales (4B, 8B), diverse datasets, and rigorous ablation studies.

**A. Core Performance vs. TTRL (Section 4.2)**
* **Significant Pass@N Gains:** EVOL-RL consistently outperforms TTRL, particularly in multi-path reasoning. For instance, **Qwen3-4B** trained on AIME24 improves **AIME25 Pass@16 from 18.5% (TTRL) to 37.9%**, effectively doubling the performance.
* **OOD Robustness:** Models trained on simple datasets (MATH-500) achieve performance on hard benchmarks (AIME) comparable to models trained directly on hard data, proving robust generalization.

**B. Robustness Across Architectures (Appendix B.1)**
* We validated EVOL-RL on a different architecture, **OctoThinker-8B-Hybrid**. While TTRL caused a performance drop on AIME24 (Pass@16 dropped from 13.9% to 11.0%), EVOL-RL improved it to **30.3%**. This confirms our method is architecture-agnostic.

**C. Comparison with Other Baselines (Appendix B.2)**
* We compared EVOL-RL against **Self-Consistency** and **Entropy Minimization**. These single-signal baselines have similar performance compared to TTRL, verifying that simple confidence/consistency signals are insufficient to stop collapse.

**D. Ablation Studies (Section 4.3 & 4.5) & Our Designed Reward on RLVR**
* **Label-Free:** The **Novelty Reward** is crucial for preventing lock-in on easy datasets (MATH-500), while **Entropy Regularization/Clipping** are vital for kickstarting exploration on hard datasets (AIME24).
* **Supervised (RLVR):** When applied to supervised learning (with ground truth), our diversity components still improved Pass@16 by 7-12%, showing universal benefit.

**E. Training Dynamics & The "Evolving Point" (Section 4.4 & Appendix B.3, B.4)**
* **Dynamics Analysis:** Unlike TTRL which flatlines at low entropy, EVOL-RL exhibits a distinct **"Evolving Point"** where entropy, response length, and accuracy simultaneously recover and rise. This pattern is consistent across **4B and 8B models** (Appendix B.4).
* **Signal Quality:** Analysis of the pseudo-labels (Appendix B.3) shows that the majority vote accuracy continuously improves under EVOL-RL, creating a positive feedback loop.

**F. Generalization to Non-Math Domains (Section 4.6)**
* EVOL-RL demonstrates strong transfer capabilities. On **MMLU-Pro** and **SuperGPQA**, it significantly outperforms TTRL (e.g., MMLU-Pro Pass@4: 78.5% vs. 73.9%), indicating the learned reasoning skills are fundamental, not just math-specific.

**G. Feasibility Validation (Appendix B.5 & B.6)**
* **Cost:** We show that the computational overhead for the embedding-based novelty reward is marginal (stable at ~100s/step) compared to the generation cost.
* **validity:** Case studies confirm that embedding cosine similarity accurately clusters reasoning paths by logic, distinguishing between different valid solutions and errors.

### **3. Theoretical Justification (Appendix D)**
We provide a rigorous mathematical proof demonstrating that:
* An objective maximizing only correctness ($J_0$) allows the optimal policy to collapse to a single deterministic solution.
* Our similarity-augmented objective ($J_{\lambda}$) guarantees that the optimal policy must distribute probability mass across **multiple correct reasoning modes**, maximizing entropy over the correct set.

We believe this work offers a fundamental step forward in autonomous LLM evolution by mathematically and empirically solving the diversity collapse problem.

Best regards,
Authors of Submission 8707

---

### Meta-Review · Area_Chair_L5K4 · 2026-01-09

**Summary:**

This paper introduces EVOL-RL, a label-free reinforcement learning framework designed to address "entropy collapse" in self-training methods for language models. The core insight is that methods relying solely on majority voting for reward cause the policy to collapse onto a single, often low-quality, solution mode. EVOL-RL introduces a dual-objective reward that balances selection (via majority voting) with variation (via a novelty reward based on semantic similarity of reasoning traces). Extensive experiments on mathematical reasoning benchmarks (e.g., AIME, MATH) demonstrate significant improvements in both Pass@1 and, crucially, Pass@N metrics, showing the method preserves and even enhances solution diversity while improving accuracy. The work includes theoretical analysis, training dynamics studies, and validation on non-mathematical domains.

**Reviewer Concerns:**

Reviewers zE94 concerns about whether the method works beyond the Qwen model family were addressed with new experiments on OctoThinker-8B (based on Llama-3.1), where EVOL-RL prevented collapse and achieved large gains over the TTRL baseline.

Reviewers zAS9 suggests a new runtime analysis confirmed the overhead of the novelty reward is minor and stable relative to the dominant cost of generation and training.

**Reviewer Scores:**

Reviewer zAS9: 4. Concerns were addressed; no follow-up score change was recorded.

Reviewer zE94: 4. The review's validity is in question due to potential misconduct. The authors' responses nonetheless addressed the technical points raised.

Reviewer UJzh: 4. Concerns were addressed; no follow-up score change was recorded.

Reviewer 5QAG: 6. The reviewer initially found the paper strong and their suggestions (model generality, baselines, cost) were thoroughly addressed.

---

### Decision · Program_Chairs · 2026-01-26

Reject